# Abundant mRNA m¹A modification in dinoflagellates: a new layer of gene regulation

Chongping Li [ID][1,2,7], Ying Li[1,2,3,4,7], Jia Guo[2,5,7], Yuci Wang[1,2,4,7], Xiaoyan Shi[1,2], Yangyi Zhang[1,2], Nan Liang[1,2], Honghui Ma[6], Jie Yuan [ID][3✉], Jiawei Xu [ID][2,5✉] & Hao Chen [ID][1,2,4,5✉]

## Abstract

**Dinoflagellates, a class of unicellular eukaryotic phytoplankton, exhibit minimal transcriptional regulation, representing a unique model for exploring gene expression. The biosynthesis, distribution, regulation, and function of mRNA N1-methyladenosine (m¹A) remain controversial due to its limited presence in typical eukaryotic mRNA. This study provides a comprehensive map of m¹A in dinoflagellate mRNA and shows that m¹A, rather than N6-methyladenosine (m⁶A), is the most prevalent internal mRNA modification in various dinoflagellate species, with an asymmetric distribution along mature transcripts. In *Amphidinium carterae*, we identify 6549 m¹A sites characterized by a non-tRNA T-loop-like sequence motif within the transcripts of 3196 genes, many of which are involved in regulating carbon and nitrogen metabolism. Enriched within 3′UTRs, dinoflagellate mRNA m¹A levels negatively correlate with translation efficiency. Nitrogen depletion further decreases mRNA m¹A levels. Our data suggest that distinctive patterns of m¹A modification might influence the expression of metabolism-related genes through translational control.**

**Keywords** N1-methyladenosine; Dinoflagellates; Post-transcriptional Regulation; mRNA Translation
**Subject Categories** RNA Biology; Translation & Protein Quality

## Introduction

Cellular RNA undergoes over 170 covalent chemical modifications, contributing to the complexity of genetic information (Wiener and Schwartz, 2021). Despite not being the most decorated RNA species, next-generation sequencing methods facilitated the transcriptome-wide mapping of various nucleotide chemical modifications, including m⁶A, pseudouridine (Ψ), 5-methylcytidine (m⁵C), m¹A, N4-acetylcytidine (ac⁴C), ribose methylations (Nm), and N7-methylguanosine (m⁷G) in messenger RNAs (mRNAs) (Roundtree et al, 2017). Of these modifications, m⁶A is one of the most prevalent internal mRNA modifications and has garnered significant attention for its pivotal role in post-transcriptional gene regulation (Murakami and Jaffrey, 2022).

As another form of adenosine methylation, m¹A was initially identified in total RNA and has been extensively investigated in non-coding RNA like ribosomal RNA (rRNA) and transfer RNA (tRNA) (Xiong et al, 2018). With methyl-adduct and positive charge at physiological conditions, m¹A can block the normal base pairing with thymidine or uridine, alter local RNA secondary structure and affect the protein-RNA interactions (Xiong et al, 2018; Zhang and Jia, 2018). The mRNA m¹A/A molar ratio ranges from 0.015% to 0.054% in mammalian cells, which is only about 10% of the level of mRNA m⁶A (Dominissini et al, 2016; Li et al, 2016). The earlier comprehensive transcriptome-wide mapping revealed that m¹A occurred widely on thousands of different gene transcripts, exhibited enrichment in their 5′-untranslated region (5′ UTR), changed dynamically in response to stress, and was correlated with translation enhancement (Dominissini et al, 2016; Li et al, 2016). However, subsequent studies demonstrated that cytosolic mRNAs were sparsely modified with m¹A in the tRNA T-loop like structures at very low stoichiometries, which led to translation repression and were invariably catalyzed by TRMT6/TRMT61A complex (Safra et al, 2017; Schwartz, 2018). Another study also reported that m¹A was a rare internal mRNA modification and only appeared in one mitochondrial transcript (Grozhik et al, 2019). Thus, due to the limited number of gene transcripts bearing m¹A sites, the biological function of these internal m¹A modifications in mammalian cells awaits further verification. In *Petunia hybrida*, a plant species containing relatively high amounts of m¹A (0.25–1.75% of total adenines in mRNA), m¹A modifications were identified in 3231 genes, mainly located in coding sequence near start coding regions, and showed dynamic distribution upon ethylene treatment (Yang et al, 2020). Very recently, Sun et.al reported that CAG repeat RNA expansions increase levels of m¹A, which alters subcellular distribution of TDP-

[1]Department of Human Cell Biology and Genetics, Joint Laboratory of Guangdong & Hong Kong Universities for Vascular Homeostasis and Diseases, School of Medicine, Southern University of Science and Technology, Shenzhen 518000, China. [2]The First Affiliated Hospital of Zhengzhou University & Institute of Reproductive Health, Henan Academy of Innovations in Medical Science, Zhengzhou 450000, China. [3]Shenzhen People's Hospital, 3046 Shennan E Rd, Shenzhen 518020, China. [4]Shenzhen Key Laboratory of Gene Regulation and Systems Biology, Southern University of Science and Technology, Shenzhen 518000, China. [5]NHC Key Laboratory of Birth Defects Prevention, Zhengzhou 450000, China. [6]Department of Liver Surgery and Transplantation, Liver Cancer Institute, Zhongshan Hospital, Institutes of Biomedical Sciences, Key Laboratory of Carcinogenesis and Cancer Invasion of Ministry of Education, Key Laboratory of Medical Epigenetics and Metabolism, Fudan University, Shanghai 200000, China. [7]These authors contributed equally: Chongping Li, Ying Li, Jia Guo, Yuci Wang. ✉E-mail: yuanjie@szhospital.com; fccxujw@zzu.edu.cn; chenh7@sustech.edu.cn

43 and consequentially contributes to neurodegeneration (Sun et al, 2023). However, the role of these m$^1$A modifications in gene expression regulation remains largely unexplored.

Dinoflagellates, consisting of important marine primary producers, harmful bloom-forming microalgae, and photo-symbionts of marine invertebrates, are a major group of unicellular eukaryotic marine phytoplankton. Their unique characteristics, including permanently condensed liquid-crystalline chromosomes lacking nucleosomes (Wong, 2019), alternating unidirectional gene arrays (Nand et al, 2021; Nelson et al, 2021), mRNA maturation with trans-splicing (Zhang et al, 2007), and non-typical promoter harboring no TATA box (Lin et al, 2015), distinguish them from other eukaryotes in terms of chromatin organization and gene expression. Instead of controlling gene expression at the transcriptional level, it was believed that dinoflagellates mainly rely on post-transcriptional regulation (Roy and Morse, 2013; Zaheri and Morse, 2022). There is a clear paucity of both basal and promoter-specific transcription factors in dinoflagellates genomes, which encode only half number of expected components of RNA polymerase II (RNAP II) and less than 10% of gene specific transcription factors compared to classical eukaryotes (Roy and Morse, 2013; Zaheri and Morse, 2022). Numerous studies also indicated that for different dinoflagellate species including *Symbiodinium* sp., *Lingulodinium* sp., *Scrippsiella* sp., and *Alexandrium* sp., only a few genes exhibited significant changes in their transcript abundance under light-dark transition, temperature change, circadian rhythm, and stressful conditions, which implies that post-transcriptional regulation might play a vital role in controlling gene expression of dinoflagellates (Chen et al, 2024; Moustafa et al, 2010; Roy et al, 2014; Yang et al, 2011; Zaheri et al, 2019).

As stated above, there is still controversy regarding the existence and role of m$^1$A in eukaryotic mRNA. Dinoflagellates lack transcription control over gene expression and primarily depend on post-transcriptional regulation. In this study, we found an unexpectedly high m$^1$A level in mRNA of dinoflagellates, making them an ideal model system to investigate the function of m$^1$A within eukaryotic mRNA. We identified thousands of m$^1$A-modified transcripts present in dinoflagellate transcriptome, which showed marked difference with reported distribution pattern in typical eukaryotes in terms of the major enrichment within 3′ UTR, and non-T loop-structure like motif. m$^1$A methylation is involved in regulating mRNA fates through decreasing translation efficiency and may be utilized to cope with nutrient stress. These findings clearly demonstrated that m$^1$A governs mRNA translation rates and adds another layer of gene expression control at post-transcriptional level in dinoflagellates.

## Results and discussion

### Dinoflagellate mRNA is highly decorated by m$^1$A modification

Due to the cholesteric liquid-crystalline structure of chromosomes, extensive use of 5′-trans-splicing in mRNA and reduced role of transcriptional regulation in dinoflagellates (Fig. 1A), we envisaged that RNA modification may be employed to post-transcriptionally regulate gene expression. Therefore, we first examined the abundance of several frequently occurred RNA modifications,

including m$^1$A, m$^6$A, m$^6$Am, and m$^7$G, in dinoflagellate *A. carterae*. The results showed that all of these RNA modifications were readily observed in total cellular RNA, while only m$^6$A and m$^1$A could be detected with considerable levels in their poly(A)+ mRNA fractions (Appendix Fig. S1A–D).

Next, we employed the ultrasensitive liquid chromatography with tandem mass spectrometry (LC-MS/MS) to measure and compare the absolute levels of m$^1$A and m$^6$A in mRNAs from HEK293T and dinoflagellates. Consistent with the very low amount of m$^1$A in mammalian cells in previous studies (Dominissini et al, 2016; Li et al, 2016), our LC-MS/MS results showed that m$^1$A level is barely detected (near to baseline) in HEK293T mRNA and m$^1$A accounts for about 1.65% of the total adenines in total RNA of HEK293T (Fig. 1B,C). Surprisingly, the m$^1$A content in *A. carterae* mRNA reached up to 3.05% of the total adenines (Fig. 1B,C), which is close to the m$^1$A level of its total RNA and two orders of magnitude higher than reported amounts of m$^1$A in mammalian cells (Dominissini et al, 2016; Li et al, 2016). Moreover, the dot blot analysis using the m$^1$A-specific antibody also demonstrated that *A. carterae* contains much more m$^1$A in their mRNA than mammalian HEK293T cell lines (Fig. 1D). In addition, our results also revealed that the m$^6$A constitutes about 0.63% of the total adenine in *A. carterae* mRNA (Fig. 1B,C), which is only equivalent to one-fifth of the corresponding m$^1$A level and slightly higher than m$^6$A amount of HEK293T mRNA (~0.35%). The relatively higher amount of m$^1$A than m$^6$A level in mRNA was also observed in other dinoflagellates including *Crypthecodinium cohnii* and *Symbiodinium* sp., while not observed in unicellular green algae *Chlamydomonas reinhardtii* (Appendix Fig. S2). Thus, this proportion of mRNA m$^1$A in dinoflagellates is likely conserved among different taxa of dinoflagellates and is much higher than other eukaryotes. Recently, it was uncovered that abundant m$^6$A resides in poly(A) tails of *Trypanosoma brucei* mRNA (Viegas et al, 2022), we asked whether dinoflagellate m$^1$A modifications are also located within mRNA poly(A) tails. Therefore, we isolated the poly(A) tail and non-poly(A) segments of *A. carterae* mRNA and measured the abundance of m$^1$A in different fractions. The results showed that these m$^1$A sites are mainly distributed in the non-poly(A) region instead of poly(A) tails of the mature transcripts (Appendix Fig. S3A,B).

### m$^1$A profile throughout the transcriptome of Dinoflagellate *A. carterae*

To gain further insight into the roles of m$^1$A, we sought to characterize m$^1$A sites at a transcriptome-wide level by using anti-m$^1$A antibody-based MeRIP (m$^1$A-specific methylated RNA immunoprecipitation). A total of 13481 peaks were detected in the mRNAs of 10794 genes in *A. carterae* under normal growth conditions (Fig. EV1A). On average, each methylated gene exhibits an average of 1.25 peaks, and the majority (76.4%) of these m$^1$A peaks are found in the stop codon segment and 3′UTR of the encoded transcripts (Fig. EV1A). To exclude the non-specific binding of the anti-m$^1$A antibody, we conducted a single base resolution analysis of m$^1$A sites in the *A. carterae* by using m$^1$A-seq-TGIRT, as described in a previous study (Safra et al, 2017). The results showed that m$^1$A-specific misincorporation events happened in a much higher ratio compared to the input control samples (Fig. EV1B). Despite using polyA-selected transcripts for

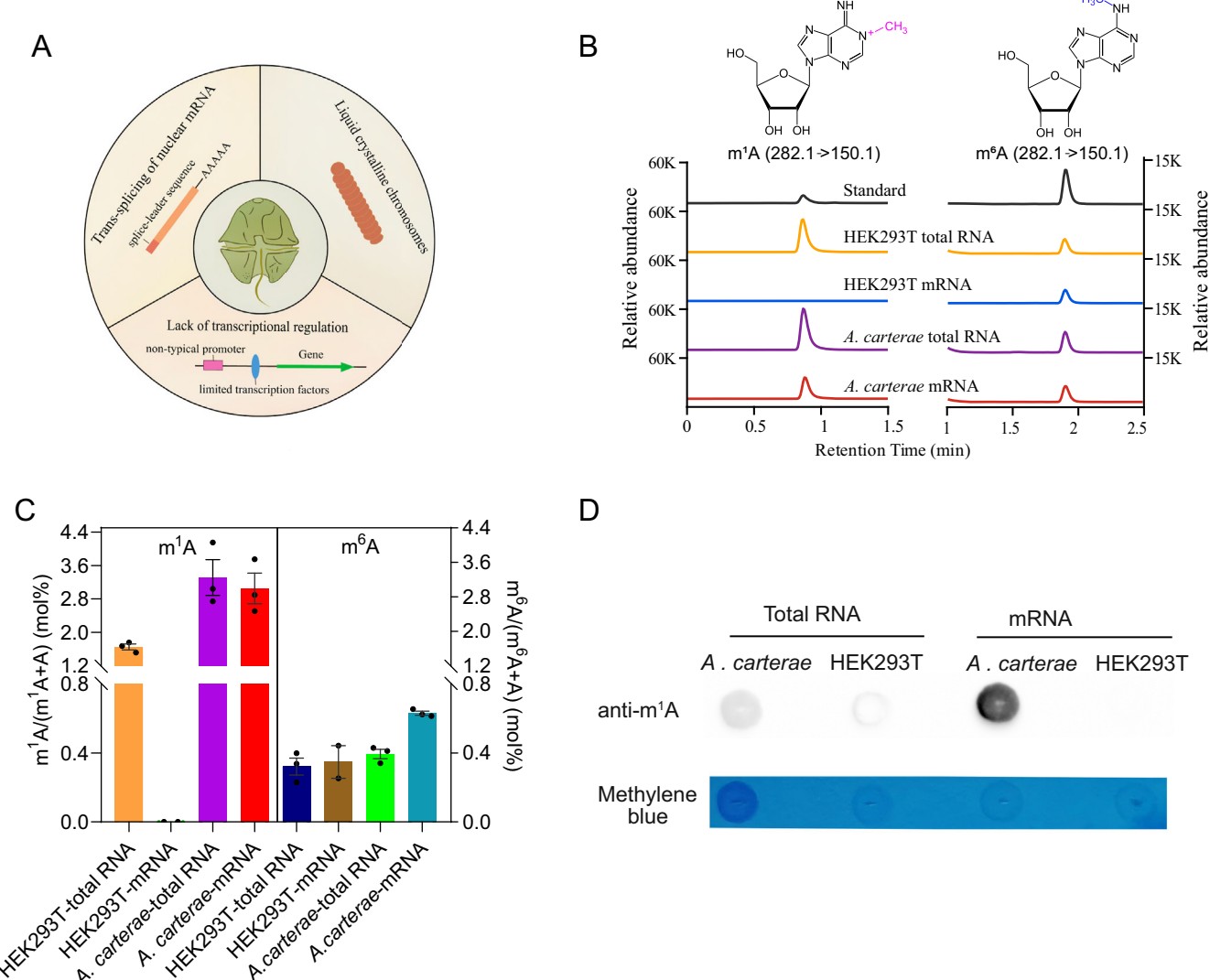

**Figure 1. Quantitative detection of m¹A levels in the dinoflagellate *A. carterae*.**

(A) Several unusual features in dinoflagellates; (B, C) LC-MS/MS chromatograms and quantification of m¹A and m⁶A levels in the mRNA and total RNA isolated from *A. carterae* and HEK 293T cell lines. Mean values ± SE are shown. Except for the two replicates used in the mRNA sample from HEK 293T cell lines, all other data were obtained with three biological replicates. (D) Dot-blot assay of m¹A abundance in both total RNA and mRNA of *A. carterae* and HEK293T cell lines, respectively. Source data are available online for this figure.

library construction, our sequencing results revealed reads mapping to the large ribosomal subunit 28S rRNA, and we observed a conserved m¹A site, present in both mammalian and yeast cells (Sharma et al, 2018), in the 28S rRNA of *A. carterae* in both the input and m¹A-seq-TGIRT samples (Fig. EV1C). This observation further confirms the reliability of our newly identified m¹A sites. In total, 6549 m¹A sites were identified in the mRNAs of 3186 genes under normal growth conditions (Fig. 2A), of which 2196 genes exhibited overlap with the MeRIP-identified peaks (Fig. EV1D). Metagene analysis was performed to identify the transcript segment(s) enriched with m¹A sites, and the results demonstrate that the 3′UTR is the region most strongly enriched in m¹A sites (Fig. 2B–D). Specifically, 81.4% of these m¹A sites are distributed in 3′UTR of the encoded transcripts, while only 14.2% and 4.4% of

these peaks are located in CDS and 5′UTR segment, respectively (Fig. 2B). Thus, in contrast to the 5′UTR enrichment pattern reported in early studies (Dominissini et al, 2016; Li et al, 2016; Yang et al, 2020), we found m¹A is mostly installed within 3′UTR of dinoflagellates mRNA.

On the other hand, analysis of the upstream 3-mer distribution frequency and sequence logo plotting of all identified m¹A sites revealed that m¹A is enriched in the NNCA sequence context (Fig. 2E). This is notably different from the m¹A motifs in mammalian cells, which resemble tRNA T-loop-like elements (GUUCNANNC) (Li et al, 2017; Safra et al, 2017). Previous studies have reported that the tRNA m¹A methyltransferase complex TRMT6/61A is responsible for the generation of the scarce cytosolic mRNA m¹A modification in mammalian cells (Li et al, 2017; Safra et al, 2017). However, we

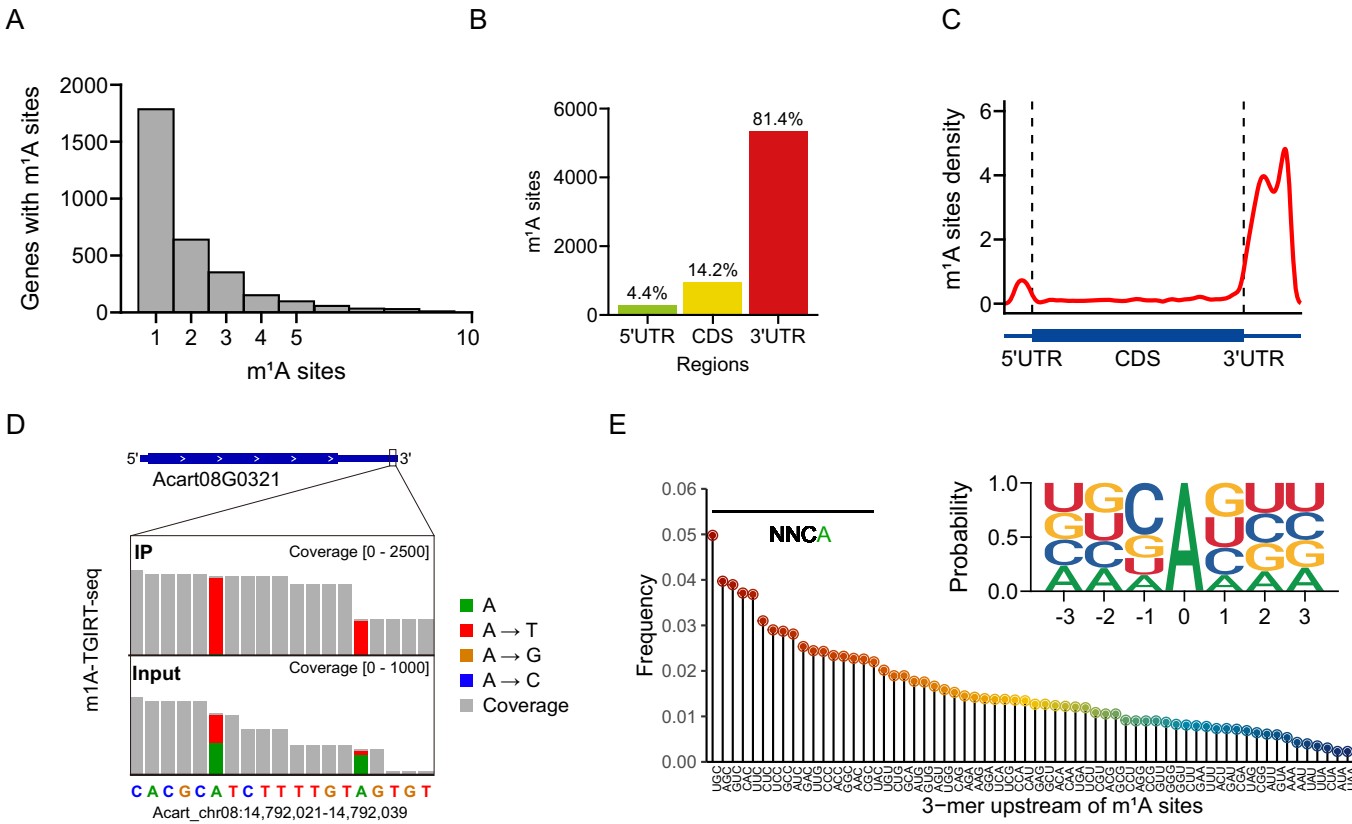

**Figure 2. The transcriptome-wide landscape of m¹A modification in *A. carterae*.**

(A) The detailed numbers of genes with m¹A-methylated transcripts exhibiting 1, 2, 3, or 4+ modified sites as identified by the m¹A-seq-TGIRT approach. (B) Percentage of m¹A sites in different regions of the mRNA. (C) Metagene profile of m¹A sites along the full-length normalized transcripts. The 5′UTR, CDS and 3′UTR were normalized to their average length according to the annotation of *A. carterae* genome. (D) Snapshot of m¹A sites-mediated misincorporation within the 3′UTR of representative transcript in *A carterae*. The y-axis represents the sequencing coverage depth. (E) The upstream 3-mer distribution frequency of all identified m¹A sites. The inset depicts the sequence logo frequency of these m¹A sites. Source data are available online for this figure.

cloned and purified *A. carterae* homologs of putative tRNA methyltransferase, the AcTRMT6/AcTRMT61A complex (Fig. EV2A). In vitro biochemical assays indicated that AcTRMT6/AcTRMT61A cannot catalyze the production of m¹A in either *A. carterae* or HEK293T mRNA (Fig. EV2B). In contrast, new m¹A generation was observed in total RNA of these cells and *A. carterae* small RNA (Fig. EV2B). These results demonstrated that dinoflagellate mRNA m¹A methylation is installed by an as-yet-unknown methyltransferase, distinct from the typical m¹A methyltransferase of eukaryotic tRNAs.

To investigate the biological functions of m¹A-methylated mRNAs, we next performed GO and KEGG enrichment analysis. GO analysis indicated that the top-ranked lists of enriched GO categories of these m¹A-decorated transcripts are correlated with cellulose catabolic process, ribonucleotide metabolic process, photosynthesis, carbohydrate catabolic process, etc. (Fig. EV1E). Meanwhile, KEGG enrichment analysis also revealed the association between m¹A-modified genes and similar pathways such as nitrogen metabolism, pyruvate metabolism, photosynthesis proteins and others (Fig. EV1F). These results implied an important role of m¹A-meidated post-transcriptional regulation in cellular carbon/nitrogen metabolism, and might represent an important mechanism of controlling critical biological processes in dinoflagellates.

## m¹A associates with translation inhibition

Bearing in mind little transcriptional regulation in dinoflagellates and interference effects of m¹A methylation on base pairing and RNA-protein interaction, we then assessed possible function of m¹A in post-transcriptional regulation within these organisms. Our analysis showed that m¹A-methylated genes in dinoflagellate *A. carterae* have significantly higher mRNA levels than the unmethylated ones (Fig. 3A). Particularly, except the 5′UTR, the transcript abundance is positively correlated with the presence of m¹A sites in both 3′UTR and CDS region (Fig. 3B). Short poly(A) tails length has been recently regarded as a conserved feature of the highly expressed genes across eukaryotes (Lima et al, 2017; Passmore and Coller, 2022). Consistent with this notion, we also found that the length of mRNA poly(A) tail is negatively correlated with mRNA abundance (Fig. EV3A) and m¹A-marked transcripts possessed shorter poly(A) tails in *A. carterae* (Fig. 3C). In addition, mRNAs with m¹A in their 3′UTR show a slight increase in the mean GC content of neighboring nucleotides, a non-uniform distribution of m¹A sites along the 3′UTR, lower global 3′UTR GC content, and longer 3′UTR lengths compared to non-methylated mRNAs (Figs. 3D and EV3B–D). Furthermore, predictive structural analysis using ViennaRNA suggested that 3′UTR in these m¹A-marked

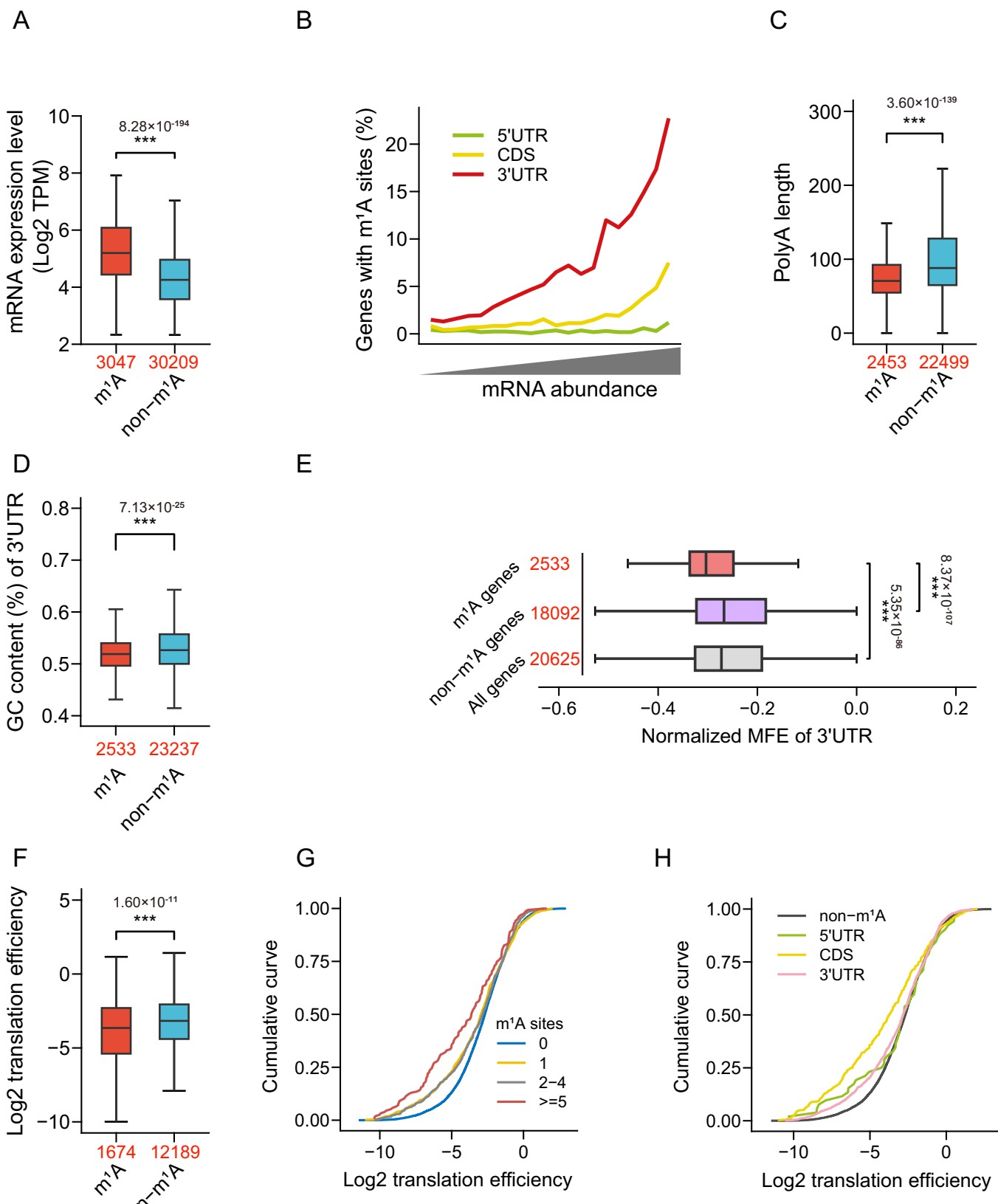

**Figure 3.  m¹A associates with mRNA abundance, gene structure, and translation efficiency.**

(**A**) The correlation of m¹A modifications with mRNA level. (**B**) Gene expression level plots against different regional methylation. (**C–F**) A comparison between the median poly(A) length (**C**), 3′UTR GC content (**D**), normalized minimum folding energy (MFE) of 3′UTR (**E**), and translation efficiency (**F**) of genes with or without m¹A modifications. (**G**) Cumulative distribution of translation efficiency of the m¹A-modified genes with different number of m¹A sites. (**H**) With same data from (**F**), translation efficiency was further compared among non-m¹A-methylated genes with m¹A-modified genes occurred in different mRNA segments. For all box plots, numbers of m¹A-modified and non-modified genes are indicated in red. Median values in each group are marked by a center line. The boxes upper and bottom boundaries denote the first quartile and third quartile, The whiskers on the plot stretch to encompass data points that fall within 1.5 times the interquartile range (IQR) from the edges of the box. $p$ values were determined using a two-sided t-test. Exact $p$-values are also reported. ***$p < 0.001$. Note: the m¹A-TGIRT-seq and Ribo-seq experiments were carried out with two independent biological replicates here ($n = 2$). Source data are available online for this figure.

genes tend to form more stable secondary structures than those in non-methylated counterparts (Fig. 3E). These results suggested that m¹A prefers to be installed on highly expressed genes and structured mRNA regions.

When localized in the CDS, m⁶A positively promotes translation rates by hindering the formation of stable secondary structures (Mao et al, 2019). Besides, 5′UTR m⁶A is also able to drive cap-independent translation in the absence of 5′ cap-binding proteins (Meyer et al, 2015). However, there is no agreement regarding the effect of m¹A on translation efficiency (Dominissini et al, 2016; Safra et al, 2017). We further explored whether m¹A methylation would affect the decoding process of endogenous transcripts. Comparative analysis of translation efficiency for transcripts with or without m¹A indicated that m¹A-modified transcripts are associated with decreased translation efficiency (Fig. 3F). Accordingly, the transcripts with more m¹A sites exhibit much lower translation efficiency (Fig. 3G). More interestingly, the significant translation inhibition is most pronounced in mRNAs with m¹A sites in their CDS regions (Fig. 3H). Thus, our results implied that the inhibitory effect of m¹A modification on transcript translation is context-dependent.

## Dynamic m¹A methylation under stress condition

Given the involvement of m¹A methylated genes in nitrogen metabolism, we further explored how m¹A methylation level and distribution respond to N-depletion stress. Similar to previous findings (Lai et al, 2011; Li et al, 2021), the growth of *A. carterae* cells was also severely inhibited under N-depletion condition (Fig. 4A). Consistent with the paucity of transcription factors and lack of differentially expressed genes at transcriptional level in many dinoflagellates (Roy and Morse, 2013; Zaheri and Morse, 2022), our RNA-seq analysis results indicated that there was only a total of 160 differentially expressed genes (DEGs) upon N-deficiency, of which 107 and 53 exhibited significant upregulation and downregulation, respectively (Fig. EV4A). The upregulated DEGs were mainly engaged in GO terms such as pigment biosynthetic process, organic cyclic compound biosynthetic process, nitrate assimilation and photosynthesis, and enriched in the KEGG pathways related to photosynthesis and beta-Alanine metabolism (Fig. EV4B,C). The downregulated DEGs are involved in GO terms associated with cellular carbohydrate biosynthetic process and response to organic substance, and KEGG pathway such as nitrogen metabolism (Fig. EV4B,C).

Recent studies suggested the control of mRNA translation in dinoflagellate *Lingulodinium polyedra* is an important post-transcriptional mechanism for circadian rhythm adaptation

(Bowazolo et al, 2022). We then utilized Ribo-seq to determine whether the translational regulation of gene expression is used to cope with N-deficiency. In contrast to quite minor changes in mRNA level, the Ribo-seq data showed that 1469 and 3037 genes showed significantly elevated and reduced translation efficiency, respectively (Fig. EV4D; Appendix Fig. S4A–D), which is consistent with the fact that nitrogen starvation represses protein synthesis (Harding et al, 2000). The genes with increased translation efficiency are mainly responsible for regulating autophagy, catabolic process, ABC transporters, lysosome, amino sugar and nucleotide sugar metabolism (Fig. EV4E,F), while the genes with decreased translation efficiency are involved in affecting purine ribonucleotide metabolic process, inorganic ion transmembrane transport, nucleotide metabolic process, photosynthesis, carbohydrate catabolic process, carbon fixation in photosynthetic organisms, and translation factors (Fig. EV4E,F). With an increase in photosynthesis at the transcriptional level but a decrease at the translational level, and a much more significant change in translation efficiency compared to mRNA levels, our data demonstrate that the dinoflagellate *A. carterae* adapts to N-depletion conditions primarily by regulating mRNA translation rates.

Various methyltransferases decorate the substrate with a methyl group transferred from S-Adenosyl methionine (SAM), the level of which is affected by nutrient availability and cellular metabolic status (Finkelstein, 1990; Obata et al, 2018). It has been shown that starvation would dramatically alter the kinetics of methyltransferases (Mentch et al, 2015; Obata et al, 2018). Therefore, we hypothesized that the deposition of m¹A in *A. carterae* mRNA is likely inhibited by N-depletion (Fig. 4B). Consistently, we found that the mRNA m¹A level significantly decreased to ~67% of that in the normal growth condition (Fig. 4C). MeRIP analysis was further conducted to define the m¹A methylome dynamics in response to nutrient stress. Under N-depletion conditions, 10569 high-confidence m¹A peaks were observed in 8854 genes, with 1879 of these genes also containing identified m¹A sites (Fig. EV5A). Differential RNA methylation region analysis revealed that among these 1879 genes, 184 regions (in 171 genes) showed significantly increased m¹A methylation, while 599 regions (in 533 genes) exhibited decreased m¹A methylation under N-deficiency (Fig. 4D,E). Notably, only 13 of these genes was differentially expressed at the transcriptional level (Fig. EV5B). In order to clarify the function of differentially methylated transcripts, GO and KEGG enrichment analyses were further conducted. With functional enrichment of the genes with decreased m¹A methylation on carbohydrate catabolic process, generation of precursor metabolites and energy, pyruvate metabolic process, polysaccharide catabolic

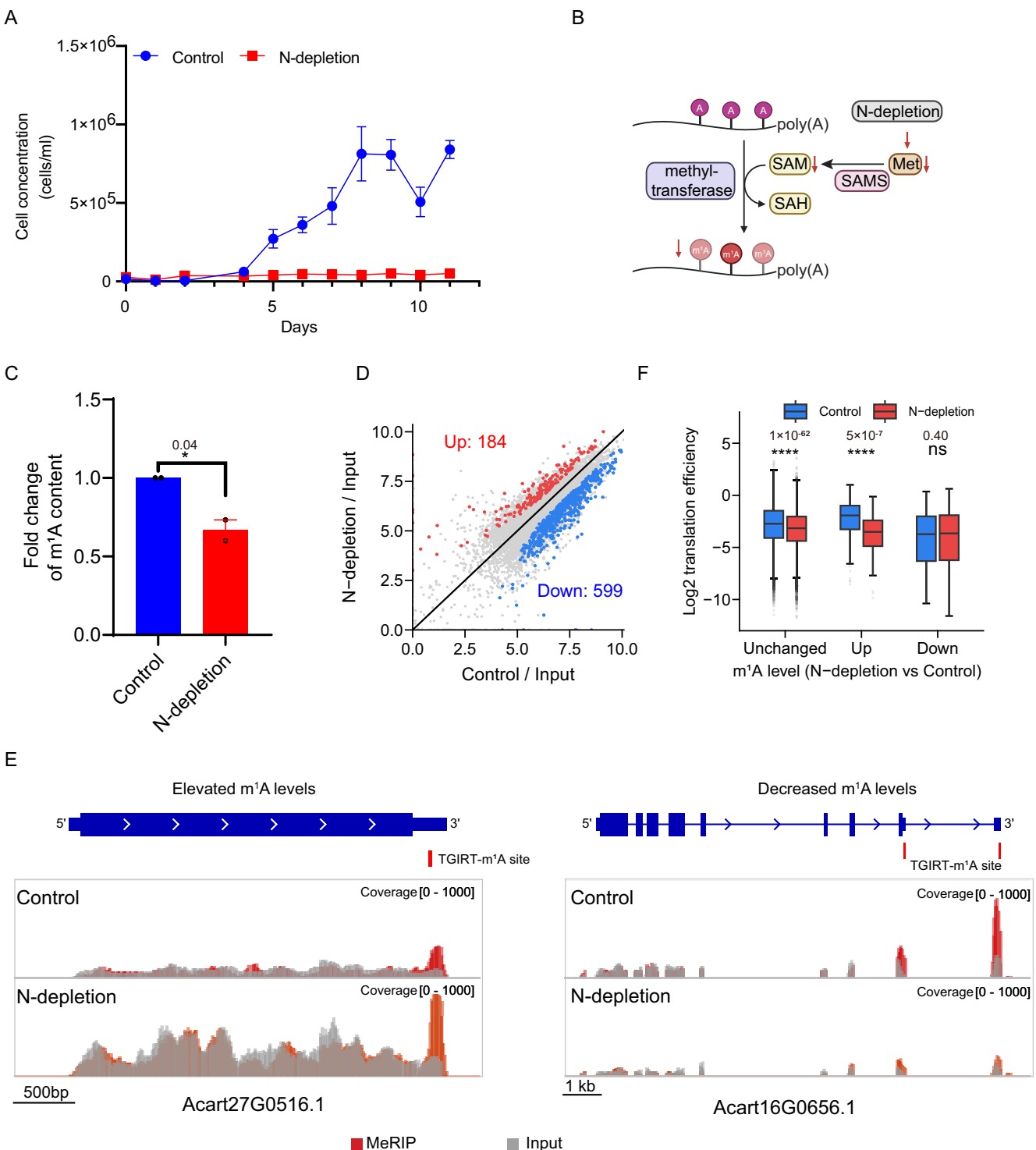

**Figure 4. Characterize the potential function of m¹A upon nitrogen deficiency.**

(**A**) Growth curves of *A. carterae* cells under different nitrogen conditions. All the data are means ± SE, $n = 3$ independent biological replicates. (**B**) Reduced supply of Met and SAM upon N-depletion likely leads to decreased m¹A decoration in mRNA. Met, SAMS, and SAH referred to methionine, S-adenosylmethionine synthase 2 and S-adenosylhomocysteine, respectively. (**C**) The fold change of mRNA m¹A level of *A. carterae* cells grown under nitrogen-depletion conditions for 6 days (with two-tailed t-test, *$p < 0.05$). All the data are means ± SE, $n = 2$ independent biological replicates. (**D**) Scatter plot of MeRIP analyses of *A. carterae* cells in N-depleted vs N-replete (Control) treatments. Red dots denote stress-upregulated peaks; blue dots denote stress-downregulated peaks. Only the differential m¹A peaks that overlapped with m¹A sites identified by m¹A-seq-TGIRT were selected as altered methylation regions. (**E**) Representative examples of elevated and reduced m¹A peaks upon N-depletion. TGIRT-m¹A site refers to m¹A sites identified by m¹A-seq-TGIRT. (**F**) Box plot shows the translation efficiency changes of genes with or without differential methylation identified in (**D**). Boxes represent 25th–75th percentile (line at the median) with whiskers at 1.5 × interquartile range (t-test (two-sided), ns, not significant; ****$p < 0.0001$). Source data are available online for this figure.

process and others (Fig. EV5C), the genes with enhanced m$^1$A methylation showed regulation on nitrogen metabolism (Fig. EV5D). These results suggested that m$^1$A modification was also utilized for post-transcriptional regulation of carbon and nitrogen metabolism in dinoflagellates.

To further characterize the regulatory roles of m$^1$A towards gene expression, we extended our analysis to compare the translation efficiency of differentially methylated mRNA under normal and N-depletion conditions. Firstly, we observed the translation efficiency of the transcripts with unchanged m$^1$A level generally decreased under nutrient stress (Figs. 4F and EV5E). This reduction primarily stems from the translational slow-down induced by starvation. However, the translation of transcripts containing m$^1$A with reduced methylation exhibited minimal sensitivity to N-depletion (Figs. 4F and EV5F), suggesting that lowering m$^1$A levels in specific transcripts may represent a viable mechanism of resisting to the global slowdown of mRNA translation. More interestingly, we identified 171 genes with significantly elevated m$^1$A levels (Fig. 4D), even though the global m$^1$A level decreased during nitrogen depletion (Fig. 4C). Importantly, augmented m$^1$A methylation levels in transcripts resulted in a more pronounced inhibition of mRNA translation, likely due to the synergistic impact of m$^1$A-induced translation repression and N-depletion-induced global translation shutdown (Figs. 4F and EV5G). Thus, the above observations showed that transcripts with diminished m$^1$A methylation demonstrated relatively higher translation efficiency, while increased m$^1$A methylation levels led to more severe translation inhibition, which is consistent with the repressive effect of m$^1$A modification on translation. Taken together, these findings underscore the notion that the abundant m$^1$A modification in mRNA functions as a repressive mark of gene expression, orchestrating regulatory control at the translational level. The translational modulation mediated by m$^1$A emerges as a potential strategy for dinoflagellates to adapt gene expression in response to environmental stimuli, such as N-depletion stress.

## Methods

### Reagents and tools table

| Reagent/Resource | Reference or Source | Identifier or Catalog number |
|---|---|---|
| **Experimental models** | | |
| *Amphidinium carterae* | Shanghai Guangyu Biotechnology Co., Ltd., Shanghai, China | GY-H35 |
| *Symbiodinium* sp. | Shanghai Guangyu Biotechnology Co., Ltd., Shanghai, China | GY-H50 |
| *Crypthecodinium cohnii* | ATCC | 30556 |
| HEK293T cells | Hao Chen Lab | |
| *Escherichia coli* BL21 (DE3) | AlpalifeBio | Cat #KTSM104L |
| **Recombinant DNA** | | |
| pETDuet-1 | Sigma-Aldrich | Cat #71146-3CN |
| pETDuet-1-AcTRMT-AcTRMT61A | This study | N/A |

| Reagent/Resource | Reference or Source | Identifier or Catalog number |
|---|---|---|
| **Antibodies** | | |
| Mouse Anti-1-methyladenosine (m1A) mAb | MBL | Cat #D345-3 |
| Goat Anti-Mouse IgG (H&L) HRP | Signalway Antibody | Cat #L35007 |
| **Oligonucleotides and other sequence-based reagents** | | |
| VAHTS RNA Adapters Set 1 - Set 2 for Illumina | Vazyme, Nanjing, China | Cat #N803 |
| VAHTS Small RNA Index Primer Kit for Illumina | Vazyme, Nanjing, China | Cat #N816 |
| 3'-RNA adapter | This study | Methods and Protocols (m$^1$A-seq-TGIRT) |
| RT primer | This study | Methods and Protocols (m$^1$A-seq-TGIRT) |
| 5'-DNA adapter | This study | Methods and Protocols (m$^1$A-seq-TGIRT) |
| PCR primers | This study | Methods and Protocols (m$^1$A-seq-TGIRT) |
| **Chemicals, enzymes, and other reagents** | | |
| Proteinase K | NEB | Cat #P8107S |
| Nuclease P1 | NEB | Cat #M0660S |
| Antarctic Phosphatase | NEB | Cat #M0289S |
| T4 Polynucleotide Kinase | NEB | Cat #M0201S |
| T4 RNA Ligase 1 | NEB | Cat #M0204S |
| Pierce Protein A/G Magnetic Beads | ThermoFisher | Cat #88803 |
| RNA Fragmentation Reagents | Invitrogen | Cat #AM8740 |
| RNase I | Invitrogen | Cat #EN0601 |
| VAHTS mRNA Capture Beads | Vazyme, Nanjing, China | Cat #N401-01 |
| RNAzol®RT | MRC | Cat #RN190 |
| EDTA-free protease inhibitor | Roche Diagnostics | Cat #04693132001 |
| RNaseOff™ RNase Inhibitor | Cwbio, Beijing, China | Cat #CW3335S |
| Triquick RNA Extraction Reagent | Shanghai Acmec Biochemical Co., Ltd, China | Cat #AC13984 |
| AMPure XP beads | Beckman Coulter | Cat #A63880 |
| TGIRT-III | HaiGene Biotech Co.,Ltd, China | Cat #D0310 |
| Ni NTA Beads 6FF | Changzhou Smart-Lifesciences Biotechnology Co., Ltd., China | Cat #SA005005 |
| Clarity Western ECL Substrate | BIO-RAD | Cat #1705061 |
| **Software** | | |
| TrimGalore v0.6.7 | https://github.com/FelixKrueger/TrimGalore | |

| Reagent/Resource | Reference or Source | Identifier or Catalog number |
|---|---|---|
| RiboseQC v0.99 | Calviello et al, 2019 | |
| STAR v2.7.10a | Dobin et al, 2012 | |
| StringTie v2.1.7 | Pertea et al, 2015 | |
| RiboDiff v0.2.2 | Zhong et al, 2016 | |
| MACS2 v2.2.7.1 | Zhang et al, 2008 | |
| DiffBind | Stark and Brown, 2012 | |
| edgeR | Robinson et al, 2010 | |
| ViennaRNA | Lorenz et al, 2011 | |
| clusterProfiler v4.2.2 | Yu et al, 2012 | |
| **Other** | | |
| Oligo Clean & Concentrator | Zymo Research | Cat #D4060 |
| RNA Clean & Concentrator-5 | Zymo Research | Cat #R1013 |
| RNA Clean & Concentrator-25 | Zymo Research | Cat #R1017 |
| ZR small-RNA PAGE Recovery Kit | Zymo Research | Cat #R1070 |
| Equalbit RNA HS Assay Kit | Vazyme, Nanjing, China | Cat #EQ211-01 |
| Equalbit 1 × dsDNA HS Assay Kit | Vazyme, Nanjing, China | Cat #EQ121-02 |
| VAHTS Universal V8 RNA-seq Library Prep Kit for Illumina | Vazyme, Nanjing, China | Cat #NR605-01 |
| VAHTS Small RNA Library Prep Kit for Illumina | Vazyme, Nanjing, China | Cat #NR811-01 |
| Countess TM II FL cell counter | ThermoFisher | |
| Qubit | ThermoFisher | |
| Illumina Novaseq 6000 | Illumina | |
| DNBSEQ-T7 | MGI Tech Co., Ltd., China | |
| PromethION | Oxford Nanopore Technologies | |
| MGISEQ-2000RS | MGI Tech Co., Ltd., China | |

## Methods and protocols

### Cell culture

Dinoflagellates *Amphidinium carterae* and *Symbiodinium* sp. were obtained from Shanghai Guangyu Biotechnology Co., Ltd. (Shanghai, China). The strains were normally cultured in f/2 medium at a constant temperature of 23 °C. The lighting conditions were maintained under a 12:12 h light-dark cycle with a photon flux density of 50 µmol·m$^{-2}$·s$^{-1}$. Sub-culturing was performed at 7- to 10-day intervals to ensure optimal growth. The other dinoflagellate *Cryptecodinium cohnii* ATCC 30556 was purchased from American Type Culture Collection (ATCC) and routinely grown in A2E6 medium at dark conditions following official instruction. HEK293T cells were cultured in Dulbecco's modified Eagle's medium (DMEM, Gibco), supplemented with 10% fetal bovine serum (FBS, BI) and 1% penicillin-streptomycin (Thermo Fisher). The cells were maintained in a humidified incubator at 37 °C with 5% $CO_2$.

For nitrogen depletion experiment in *A. carterae*, the exponentially grown cells were washed once with sterile sea water and then cultured in normal and nitrogen-depletion f/2 medium, respectively, at an initial concentration of $2.5 \times 10^4$ cells/ml. Cell numbers of *A. carterae* cells during the whole growth period were counted using automated Countess TM II FL cell counter (Thermo Fisher). *A. carterae* cells grown at exponential stage (day 6) were used to perform mRNA isolation and m$^1$A content quantification.

### RNA preparation for LC-MS/MS analysis

Total RNA was extracted from *A. carterae*, *Symbiodinium* sp., *C. cohnii* and HEK293T cell pellets using RNAzol®RT (MRC) according to the manufacturer's protocols. The total RNA from *Chlamydomonas reinhardtii* was received as a kind gift from Prof. Hongjie Shen, Fudan University. All materials and reagents used were RNase-free to minimize RNA degradation. Polyadenylated RNA (poly(A) + RNA) was enriched from the total RNA samples by using two successive rounds of purification with VAHTS mRNA Capture Beads (Vazyme Biotech, Nanjing, China). RNA samples were digested by 50 units of nuclease P1 (NEB) in a 25-µL reaction mixture containing nuclease P1 buffer at 37 °C for 2 h, followed by addition of 5 units Antarctic Phosphatase (NEB) and 2.5 µL 10 × Antarctic Phosphatase buffer. The enzymatic digestion was further conducted at 37 °C overnight. The enzymatic reaction was terminated by adding 75 µL of methanol. The samples were then centrifuged at 13,000 rpm for 15 min. The supernatant was lyophilized using a vacuum freeze-dryer to concentrate the RNA digest and reconstituted in 50 µL of RNase-free water. 5 µL of samples was injected into LC–MS/MS system (Agilent6470 triple-quadrupole mass spectrometer or Q Exactive high-resolution benchtop quadrupole Orbitrap mass spectrometer) on an Agilent XDB-C18 column. Nucleosides were detected by using retention time and nucleoside to base ion mass transitions of $m/z$ 268.1 to 136.1 (A), $m/z$ 282.1 to 150.1, $m/z$ 282.1 to 150.1 (m$^6$A), $m/z$ 285.1 to 153.1 (d3-m$^1$A), $m/z$ 298.1 to 166.1 (m$^7$G) and $m/z$ 296.1 to 150.1 (m$^6$Am). Nucleoside standards with serial dilution (A, from 50 nM to 1600 nM; m$^1$A, from 7.5 nM to 240 nM; m$^6$A, 2 nM to 64 nM) were run in parallel to produce the standard curves and obtain the target nucleosides concentration.

### Dot blot assay

The concentration of isolated total RNA and mRNA from both *A. carterae* and HEK293T cells were firstly determined using Equalbit RNA HS Assay Kit (Vazyme Biotech, Nanjing, China) and Qubit instrument (ThermoFisher). An aliquot of each RNA sample (~100 ng) was spotted onto a positively-charged nylon membrane and subsequently crosslinked with UV light using the chamber of SG linker. The membrane was then stained with methylene blue solution (0.2% methylene blue in 0.4 M sodium acetate and 0.4 M acetic acid) to ensure equal loading. After washing, the membrane was blocked with 5% milk in 1 × TBST at room temperature for 1–2 h and incubated at 4 °C overnight with a 1:1000 dilution of anti-m$^1$A antibody (MBL #D345-3). On the second day, the membrane was washed with 1 × TBST for three times and then incubated in a 1:5000 dilution of goat anti-mouse IgG secondary antibody (Signalway Antibody LLC, Maryland, USA) for 1 h at room temperature. After washing another three times in 1 × TBST,

the blots were visualized with the Clarity Western ECL Substrate (BIO-RAD).

## MeRIP

Poly(A) + RNA of *A. carterae* was isolated using VAHTS mRNA Capture Beads (Vazyme Biotech, Nanjing, China), fragmented at 70 °C for 10 min using RNA fragmentation reagent (Invitrogen) in a 10 μL reaction mixture and purified with the RNA Clean & Concentrator-5 kit (Zymo Research). For MeRIP, Protein A/G Dynabeads (10 μL) were washed three times with 1 × IP buffer (150 mM NaCl, 0.1% NP-40, 10 mM Tris–HCl, pH 7.4), resuspended in 200 μL of 1 × IP buffer, and incubated with 1 μL of anti-m¹A antibody (MBL #D345-3) and 2.5 μL of protease inhibitor at 4 °C for one hour. The antibody-coated beads were washed twice with 1 × IP buffer and then mixed with 200 μL of IP buffer, 2.5 μL of complete EDTA-free protease inhibitor (Roche Diagnostics), 1 μL of RNase inhibitor (Cwbio, Beijing, China), and 200 ng of fragmented mRNA. 10 μL of the resultant mixture was retained as input sample. The mixture was incubated at 4 °C with gentle rotation for 3 h and washed sequentially with 1 mL of IP buffer, low-salt buffer (50 mM NaCl, 0.1% NP-40, 10 mM Tris–HCl, pH 7.4), and high-salt buffer (500 mM NaCl, 0.1% NP-40, 10 mM Tris–HCl, pH 7.4) at 4 °C for five minutes. The immunoprecipitated products were eluted using 5 units of Proteinase K in a buffer containing 5 mM Tris–HCl (pH 7.5), 1 mM EDTA, and 0.25% SDS at 37 °C for 1.5 h, then purified with the Zymo Oligo Clean & Concentrator kit (Zymo Research) and prepared for library construction using the VAHTS Universal V8 RNA-seq Library Prep Kit for Illumina (Vazyme Biotech, Nanjing, China). Both input and IP samples were sequenced by Genewiz (Jiangsu, China) on an Illumina Novaseq 6000 platform with paired-end 150 bp mode.

## m¹A-seq-TGIRT

For m¹A-seq-TGIRT, 25 μL of prewashed Protein A/G Dynabeads were incubated with 3 μL of anti-m¹A antibody (MBL #D345-3) at 4 °C for 2 h. Subsequently, 500 ng of fragmented mRNA was used for immunoprecipitation (IP), with 50 ng of purified mRNA taken as input. The IP protocol followed the same steps as described above. After digestion with Proteinase K, the RNA was purified using RNA Extraction Reagent (Shanghai Acmec Biochemical Co., Ltd, China) and then subjected to library construction as previously (Safra et al, 2017). Briefly, the purified mRNA was treated with Antarctic Phosphatase (NEB) and T4 PNK (NEB) at 37 °C for 30 min, followed by purification using the Oligo Clean & Concentrator kit (Zymo Research). The dephosphorylated RNA was then ligated to the 3′-RNA adapter (/5′Phosphate/AGAUCG-GAAGAGCGUCGUG/ddC) using T4 RNA Ligase I (NEB) at 23 °C for two hours, purified by Oligo Clean & Concentrator again, and reverse transcribed using TGIRT-III enzyme (HaiGene Biotech Co.,Ltd, China, D0310) with the RT primer (5′-ACAC-GACGCTCTTCCGA-3′) in RT reaction buffer (50 mM Tris–HCl pH 8.3, 75 mM KCl, and 3 mM MgCl₂) at 57 °C for 2 h. The reaction was terminated with 25 mM EDTA and RNA was degraded with 150 mM NaOH, heated to 70 °C for 12 min, and purified with Oligo Clean & Concentrator kit. The 5′-DNA adapter (/5′Phosphate/AGATCGGAAGAGCACACGTCTG/ddC) was ligated using T4 RNA ligase I (NEB) at 23 °C overnight. The single-stranded cDNA was then amplified by PCR using specific primers (5′-AATGATACGGCGACCACCGAGATCTACACTCTT TCCCTACACGACGCTCTTCCGATCT-3′ and 5′-CAAGCAGAA

GACGGCATACGAGATNNNNNNGTGACTGGAGTTCAGACG TGTGCTCTTCCGATCT-3′, NNNNNN denotes the 6-bp barcode sequence), and the product was purified with AMPure XP beads (0.9×). Both input and IP samples were sequenced by GenePlus Inc. (Shenzhen, China) using the DNBSEQ-T7 platform with paired-end 150 bp reads.

## RNA-seq and Ribo-seq experiments

For RNA-seq, the isolated total RNA from *A. carterae* cells (three biological replicates for all treatments) was sent to the Beijing Genomics Institute (BGI, Shenzhen, China) for strand-specific mRNA library construction and sequencing on DNBSEQ PE150 platform. On the other hand, to obtain the poly(A) tail length of *A. carteae* genes, one aliquot total RNA sample was sent to Wuhan Benagen Technology Co., Ltd. for nanopore direct RNA-sequencing using PromethION sequencer (Oxford Nanopore Technologies, Oxford, UK).

For Ribo-seq, *A. carterae* cells (~40 million cells, two biological samples for each treatment) collected in the morning were treated with 100 μg/ml cycloheximide (CHX) for 8 min. After treatment, cells were centrifuged at 2000 rpm for 10 min at 4 °C. The resulting cell pellets were washed once with CHX-containing 1 × PBS and resuspended in 2–3 volumes of lysis buffer composed of 10 mM Tris–HCl (pH 7.5), 100 mM KCl, 5 mM MgCl₂, 1% Triton X-100, 100 μg/ml CHX, 2 mM DTT, and 1 × complete EDTA-free protease inhibitor (Roche Diagnostics), followed by sonication at a power of 100 W using an ultrasonic processor (SCIENTZ-950E, SCIENTZ, China) for 2 min (3 s on/3 s off) in a ice-water bath. The supernatant was collected after centrifugation at 12,000 × *g* for 10 min at 4 °C. The optical density (OD) of the cell lysate was determined from the absorbance of the supernatant at 260 nm with NanoDrop 2000. For Ribosome Protected Fragments (RPFs) generation, 15 U of RNase I (Invitrogen) was added per 1 OD of lysate, followed by a 40-minute incubation at 22 °C with gentle mixing. RPFs were isolated using MicroSpin S-400 columns (Cytiva) and purified using the RNA Clean & Concentrator-25 kit (Zymo Research). After size-separation of RPFs using a 15% polyacrylamide TBE-urea gel (Coolaber Technology Co., Ltd, Beijing, China). Gel slices containing fragments of 25 to 34 nucleotides were excised using a scalpel. RNA recovery from the gel slices was performed using the ZR small-RNA PAGE Recovery Kit (Zymo Research). Purified RNA was mixed with 10 × T4 polynucleotide kinase reaction Buffer and T4 polynucleotide kinase and incubated at 37 °C for 20 min with gentle mixing, followed by addition of 1 × T4 DNA ligase buffer and another incubation at 37 °C for 20 min. The samples were repurified using the RNA Clean & Concentrator-25 kit (Zymo Research) and proceeded directly to cDNA library construction using the VAHTSTM Small RNA Library Prep Kit for Illumina (Vazyme Biotech, Nanjing, China). Sequencing was performed on the BGI MGISEQ-2000 platform using SE50 strategy.

## Isolation of poly(A) tail and non-poly(A) segments from *A. carterae* poly(A)+RNA

200 ng of purified poly(A) + RNA from *A. carterae* was first fragmented at 70 °C for 10 min using 10 × RNA fragmentation reagent (Invitrogen) in a 10 μL reaction mixture. Following fragmentation, the poly(A) tails were captured using VAHTS mRNA Capture Beads (Vazyme Biotech, Nanjing, China).

Meanwhile, the non-poly(A) RNA segments in the flow-through were purified using the RNA Clean & Concentrator-5 kit (Zymo Research). The resulting purified poly(A) tails and non-poly(A) segments were then analyzed using an anti-m$^1$A antibody-based dot-blot assay.

### Protein purification

The encoding sequences of both AcTRMT6 and AcTRMT61A were optimized with *Escherichia coli* codons, synthesized by Tsingke Biological Technology Co., Ltd. (Beijing, China) and inserted into the expression vector pETDuet-1 to generate pETDuet-1-AcTRMT-AcTRMT61A plasmid, which was then transformed into *E. coli* BL21 (DE3) competent cells. Cells were cultured at 37 °C until OD600 reached 0.6–0.8, then induced with 0.1 mM IPTG and incubated at 16 °C for 16 h. The harvested cells were lysed using high-pressure homogenizer (AH-NANO, ATS, China) with a pressure of 800–900 bar for four cycles in the lysis buffer (20 mM Tris–HCl, pH 7.4, 300 mM NaCl, 15 mM imidazole, 0.25% TritonX-100, 10% glycerol). The lysate was centrifuged at 20,000 rpm for 30 min, and the supernatant containing 6 × His-tagged AcTRMT-AcTRMT61A complex was purified using Ni-NTA beads (Changzhou Smart-Lifesciences Biotechnology Co., Ltd., China) following the manufacturer's guidelines. In brief, the protein-bound beads were washed three times with wash buffer (20 mM Tris–HCl, pH 7.4, 300 mM NaCl, 25 mM imidazole, 10% glycerol) and then eluted using the elution buffer (50 mM Tris–HCl, pH 7.4, 150 mM NaCl, 250 mM imidazole and 10% glycerol). Eluted proteins were pooled and dialyzed extensively with dialysis buffer (20 mM Tris–HCl pH 7.4, 50 mM NaCl, 0.5 mM EDTA, 1 mM DTT, and 10% glycerol). All purification steps were conducted at 4 °C. The purity and size of the isolated protein complex were confirmed through Coomassie blue staining of sodium dodecyl sulfate-polyacrylamide gels.

### Biochemical assay of AcTRMT6-AcTRMT61A heterocomplex activity in vitro

Biochemical assays of the purified AcTRMT6-AcTRMT61A enzymes were conducted in 50 µL reaction mixtures at 28 °C overnight. Each mixture contained 1× MTase buffer (100 mM Tris, pH 8.0, 1.0 mM DTT, 0.1 mM EDTA, 10 mM MgCl$_2$, 20 mM NH$_4$Cl), 32 µM d3-SAM (deuterium-labeled S-Adenosylmethionine), 5 µM recombinant protein complexes, and various substrates. Tested substrates included total RNA, mRNA, and small RNA from *A. carterae*, as well as total RNA and mRNA from HEK293T cells. The small RNA (<200 nt) from *A. carterae* was purified using the RNA Clean & Concentrator-25 kit (Zymo Research). For control, enzyme-inactivation reactions through heat treatment were set up using total RNA from *A. carterae* cells. After overnight incubation, the reaction mixtures were heat-inactivated at 95 °C for 5 min and cooled down in an ice bath for another 5 min. Nuclease P1 (NEB) and Antarctic Phosphatase (NEB) were then added to digest the samples for subsequent LC-MS/MS analysis.

### Bioinformatics analyses of RNA-seq, Ribo-seq, MeRIP-seq, and m$^1$A-seq-TGIRT data

A de novo genome assembly of *A. carterae* (Acart.genome.v1) based on HiFi reads of Pacific Biosciences Sequel II system was used as the reference genome in this study (Li et al, 2023). Raw reads data of RNA-seq, Ribo-seq, methylated RNA immunoprecipitation sequencing (MeRIP-seq) and m$^1$A-seq-TGIRT were prepared according to the procedure describe previously (Chothani et al, 2022). Briefly, TrimGalore (version 0.6.7) was used to remove adapters and low-quality base (-q 25 -e 0.1 --stringency 3) at first. Of note, for Ribo-seq down-stream analysis, reads with length in the range of 20–35 bp were kept. RPFs in Ribo-seq represents actively translated mRNA transcript and thus reads mapping to rRNA sequences were discarded. The trinucleotide periodicity of ribosomes was examined using RiboseQC package (v0.99) (Calviello et al, 2019). Pre-processed reads from RNA-seq, Ribo-seq, MeRIP-seq, and m$^1$A-seq-TGIRT were then aligned to Acart.genome.v1 using software STAR (version 2.7.10a) (Dobin et al, 2012). The alignment allowed a maximum of two mismatches for RNA-seq, Ribo-seq, and MeRIP-seq, while m$^1$A-seq-TGIRT allowed up to three mismatches per read, ensuring that only the best alignment for each read was retained. Note that Ribo-Seq reads were mapped with additional parameters: --outFilterScoreMinOverLread 0.33, --outFilterMatchNminOverLread 0.33.

The sorted mapping files of Ribo-seq and RNA-seq were used for quantification of with StringTie (v2.1.7) (Pertea et al, 2015). Next, the obtained read counts were converted to TPM (transcripts per million). Genes with cutoff values of TPM > 5 for RNA-Seq were used for downstream analysis. The R package DESeq2 was used to perform differential gene expression of RNA-seq datasets. Next, translational efficiency (TE) was calculated by dividing the normalized RPF counts, using a Ribo-seq TPM value larger than 0, by mRNA abundance (TE = normalized RPF counts/mRNA abundance) (Chothani et al, 2022). The differential translation efficiency gene (DTEG) was analyzed by RiboDiff (v0.2.2) (Zhong et al, 2016).

After aligning the MeRIP dataset, m$^1$A peaks enriched via IP over input control were detected using the MACS2 (v2.2.7.1) (Zhang et al, 2008) algorithm, employing a -q value threshold of 0.05 and a genome size parameter (-g) of 1.2e9, based on the combined biological replicates. Differential methylation regions with a fold change of >1.5 and p value of <0.05 were identified using DiffBind (Stark and Brown, 2012) with edgeR (Robinson et al, 2010). Following alignment for m$^1$A-seq-TGIRT dataset, samtools was used with the parameters 'mpileup -d 100000 -a' to generate a pileup file that provided detailed information about the coverage and mismatch count for each base in the reference genome. From this file, the coverage and mismatch data for each base located within exon regions were extracted. For each adenosine site within annotated transcripts in each sample, the data were filtered to retain only those sites with coverage exceeding 10 reads and at least 2 mismatches with a mismatch rate greater than 10%. These criteria identified candidate m$^1$A sites, which were then pooled for further analysis. To identify putative m$^1$A sites, we compared the coverage and mismatch rates between IP and input samples using a proportion Z-test with the 'greater' parameter (IP vs input). Sites were considered putative m$^1$A sites if they met the following criteria: p-value < 0.05, coverage greater than 10 in both IP and input samples, an IP mismatch count greater than 2, and an IP mismatch count higher than that in the input sample. Minimum free energy (MFE) calculation of m$^1$A methylated genes was performed by using the ViennaRNA package (Lorenz et al, 2011). All GO (Gene Ontology) terms and KEGG (Kyoto Encyclopedia of Genes and Genomes) pathway enrichment analyses were

performed by R package clusterProfiler (v4.2.2) (Yu et al, 2012). Finally, genes with $m^1A$ peaks intersected with the regions flanking 100 nucleotides of identified $m^1A$ sites were classified as shared methylated genes between MeRIP and $m^1A$-seq-TGIRT.

## Data availability

The raw data of this study is available in the GEO database under the accession number: GSE246953.

The source data of this paper are collected in the following database record: biostudies:S-SCDT-10_1038-S44319-024-00234-2.

## Peer review information

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

## Acknowledgements

We would like to thank Ms. Hui Zhou for her help in recombinant protein purification. The authors would also like to acknowledge the technical support from Hua Li and Lin Lin at SUSTech CRFT. This work was supported by Center for Computational Science and Engineering at Southern University of Science and Technology. This work was supported by National Key Research and Development Program of China (Grant Nos. 2022YFC2702705, 2019YFA0110900, and 2019YFA0802200), National Natural Science Foundation of China (Grant Nos. 32170819 and 32170604). This work was also supported by Pearl River Recruitment Program of Talents (2021QN02Y122) and Health Department of Guangdong Province (B2021032) to HC, Shenzhen Key Laboratory of Gene Regulation and Systems Biology (Grant No. ZDSYS20200811144002008) and Shenzhen Science and Technology Program (20231120115406001) from Shenzhen Innovation Committee of Science and Technology and Funding for Scientific Research, the Scientific and Technological Innovation Team Project of Universities (Grant No. 24IRTSTHN037) and the Joint Fund for the Cultivation of Superior Disciplines of Henan Province (Grant No. 222301420013), the Medical Appropriate Technology Promotion project, the Young and Middle-aged Academic Leaders and the Leading Talents of Henan Health Commission (to JWX), and Funding for Scientific Research and Innovation Team of The First Affiliated Hospital of Zhengzhou University (ZYCXTD2023004).

## Author contributions

**Chongping Li**: Conceptualization; Formal analysis; Investigation; Visualization; Methodology; Writing—original draft; Writing—review and editing. **Ying Li**: Formal analysis; Investigation; Visualization; Methodology; Writing—original draft; Writing—review and editing. **Jia Guo**: Investigation; Methodology; Writing—review and editing. **Yuci Wang**: Data curation; Software; Formal analysis; Visualization; Methodology; Writing—original draft; Writing—review and editing. **Xiaoyan Shi**: Methodology. **Yangyi Zhang**: Methodology. **Nan Liang**: Investigation; Methodology. **Honghui Ma**: Methodology. **Jie Yuan**: Supervision. **Jiawei Xu**: Supervision; Funding acquisition; Writing—review and editing. **Hao Chen**: Conceptualization; Data curation; Formal analysis; Supervision; Funding acquisition; Writing—original draft; Project administration; Writing—review and editing.

Source data underlying figure panels in this paper may have individual authorship assigned. Where available, figure panel/source data authorship is listed in the following database record: biostudies:S-SCDT-10_1038-S44319-024-00234-2.

## Disclosure and competing interests statement

The authors declare no competing interests.

# Expanded View Figures

**Figure EV1.  Characterization of the distribution of m¹A in the dinoflagellate mRNA.**

(**A**) Counts of MeRIP-identified m¹A peaks in different regions of the mRNA. When a peak overlaps with either the start codon or stop codon, we assigned these peaks to the "start codon" and "stop codon" regions, respectively. Otherwise, these peaks were considered to localize in the 5′UTR, CDS and 3′UTR according to their positions. (**B**) Comparison of misincorporation rates for all identified m¹A sites in IP samples and input using the m¹A-TGIRT-seq approach. (**C**) Snapshot of m¹A-mediated misincorporation detected by m¹A-TGIRT-seq within the 28s rRNA in *A carterae*. This m¹A site is known as m¹A$_{1322}$ in the 28s rRNA of mammalian and yeast cells. The y-axis represents the sequencing coverage depth. (**D**) The Venn diagram showing the number of overlapping genes identified by both m¹A-seq-TGIRT and MeRIP under normal growth conditions. (**E, F**) GO and KEGG pathway enrichment analysis of m¹A-methylated genes identified by m¹A-TGIRT-seq. *p*-values were obtained using the hypergeometric test.  Source data are available online for this figure.

▶

    

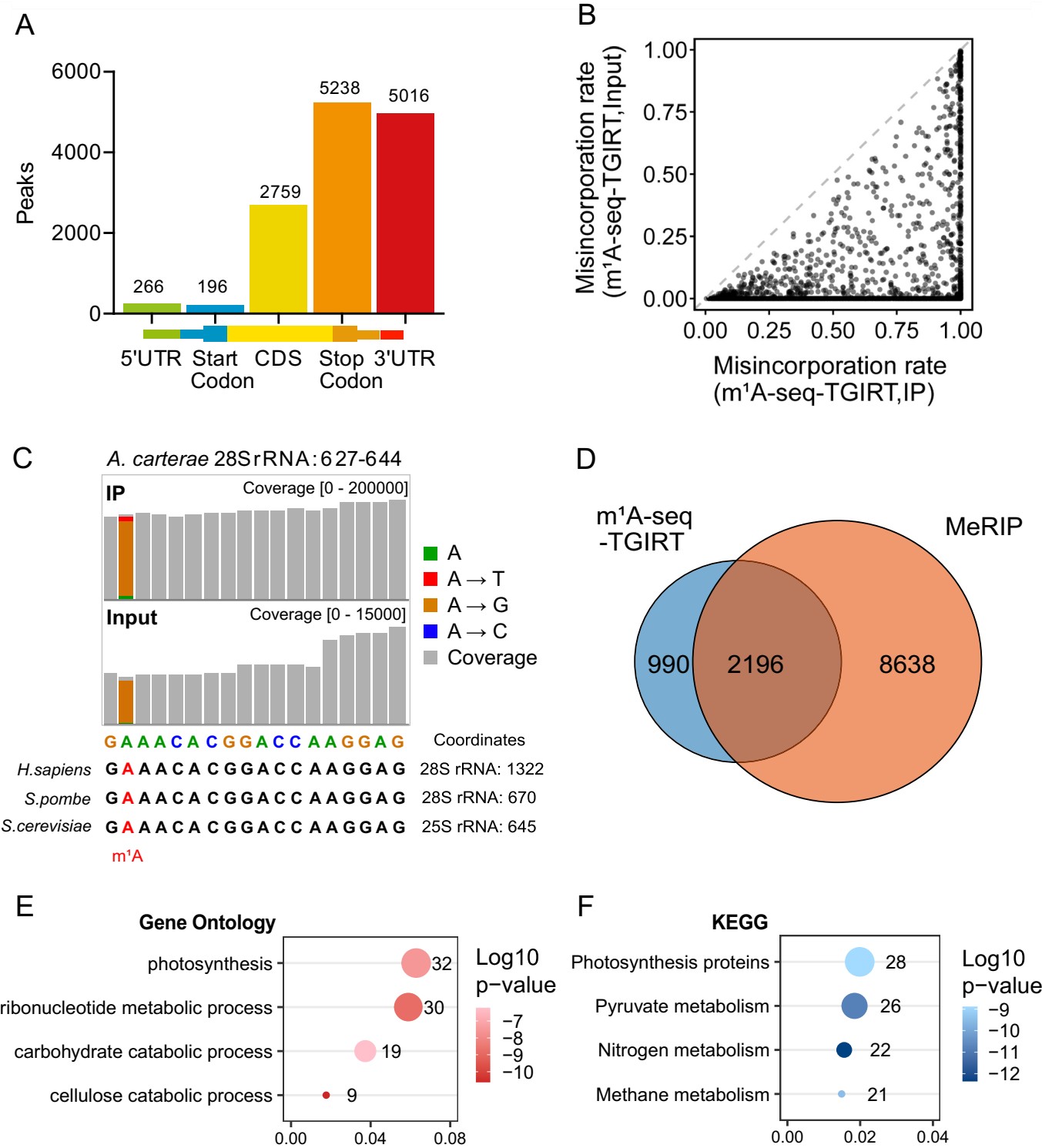

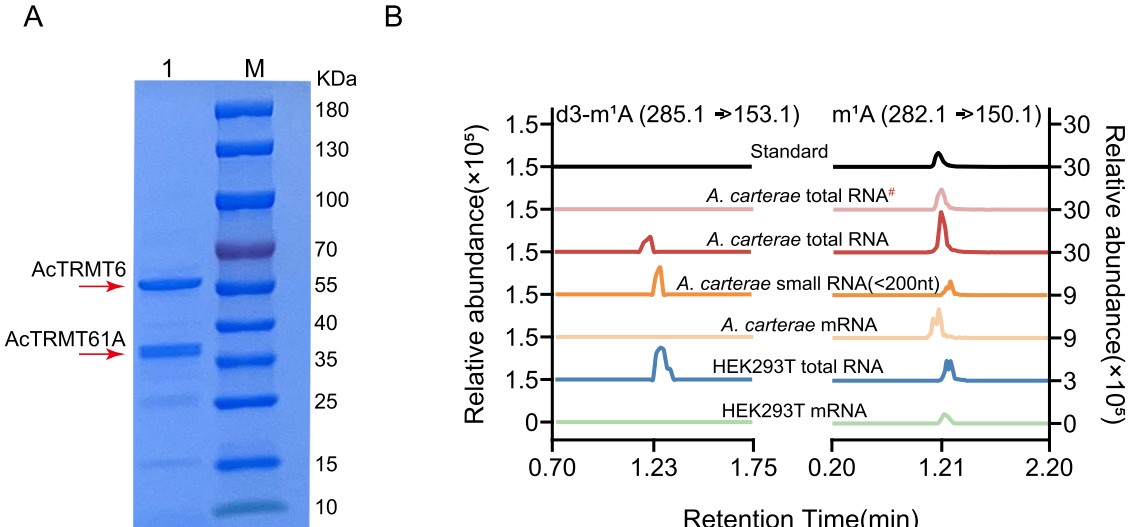

**Figure EV2. The methyltransferase activities of AcTRMT6/AcTRMT61A heterocomplex to different RNA substrates.**

(**A**) SDS-PAGE analysis showing the purified AcTRMT6/AcTRMT61A complex with expected molecular weights of 38.5 kDa (AcTRMT61A) and 51.5 kDa (AcTRMT6), respectively. Lane 1, the purified recombinant AcTRMT6/AcTRMT61A heterocomplex; Lane M, the Thermo Scientific PageRuler Plus Prestained Protein Ladder. (**B**) LC-MS/MS spectra of various RNA substrates from in vitro methylation reaction. #, represents the addition of heat-inactivated AcTRMT6/AcTRMT61A heterocomplex in this reaction mixture. The native recombinant protein complex AcTRMT6/AcTRMT61A were used in all other reactions.

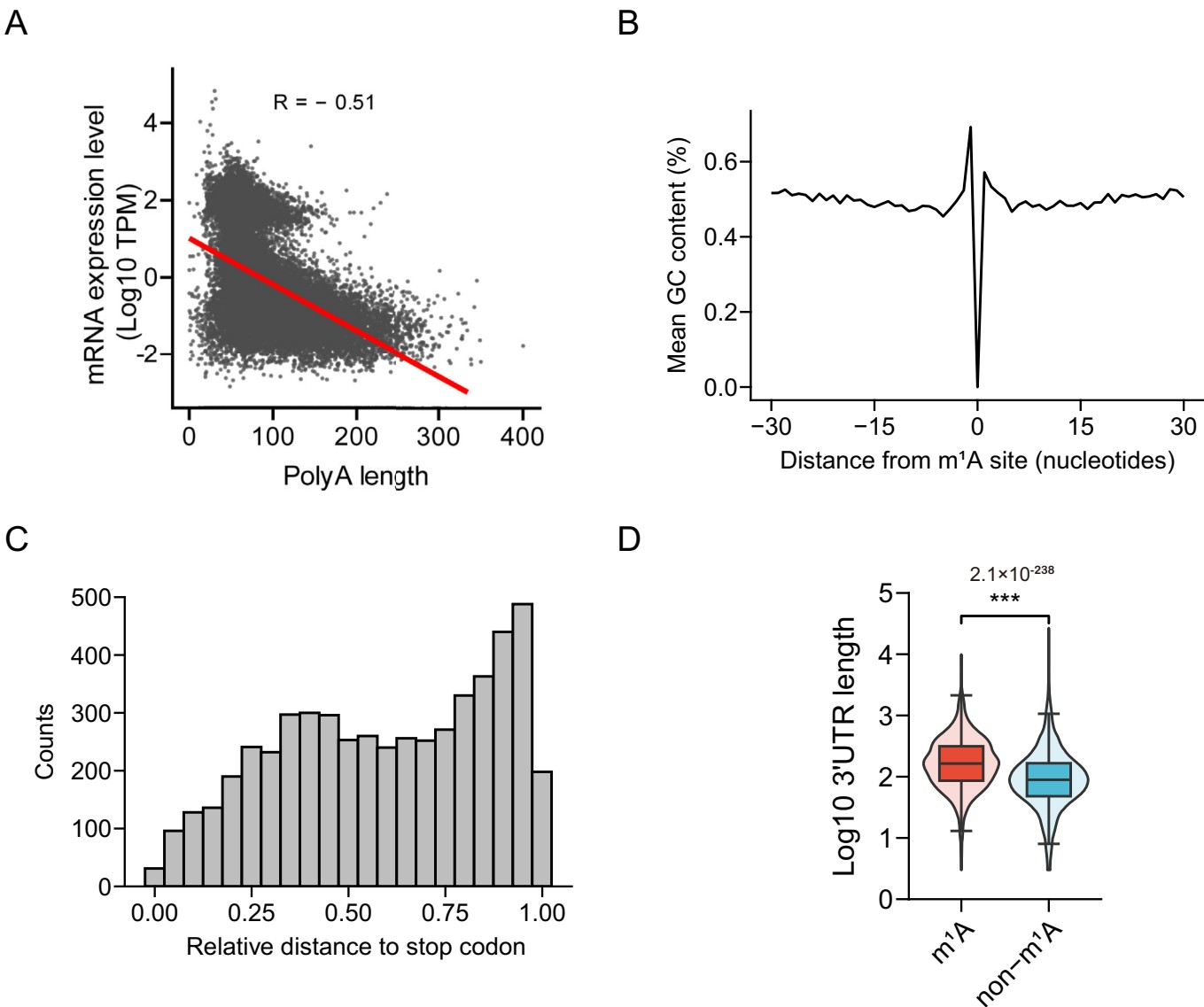

**Figure EV3. The correlation between gene expression level and poly(A) tails, and detailed characteristics of m¹A-methylated transcripts.**

(A) Correlation analysis of mRNA abundance with the poly(A) length of encoded transcripts in *A. carterae* (Pearson correlation test, $p < 2.2 \times 10^{-16}$). The distribution properties of m¹A sites within the 3'UTR (same sites used in Main Fig. 3) are shown in following Figures (**B–D**). (**B**) The mean GC content of neighboring nucleotides around all identified m¹A sites ($n = 5297$) within the 3'UTR. (**C**) The histogram showing the distribution pattern of the relative distance to the stop codon for all identified m¹A sites within the 3'UTR. (**D**) A comparison of 3'UTRs' length of genes with m¹A-methylated 3'UTRs and those with unmethylated ones (t-test (two-sided), ***$p < 0.001$). The median value in each group is indicated by a center line, with the box representing the upper and lower quartiles, and whiskers indicating the 1.5× interquartile range. Note: For EV3B–D, the m¹A-TGIRT-seq experiment was carried out with two independent biological replicates ($n = 2$). Source data are available online for this figure.

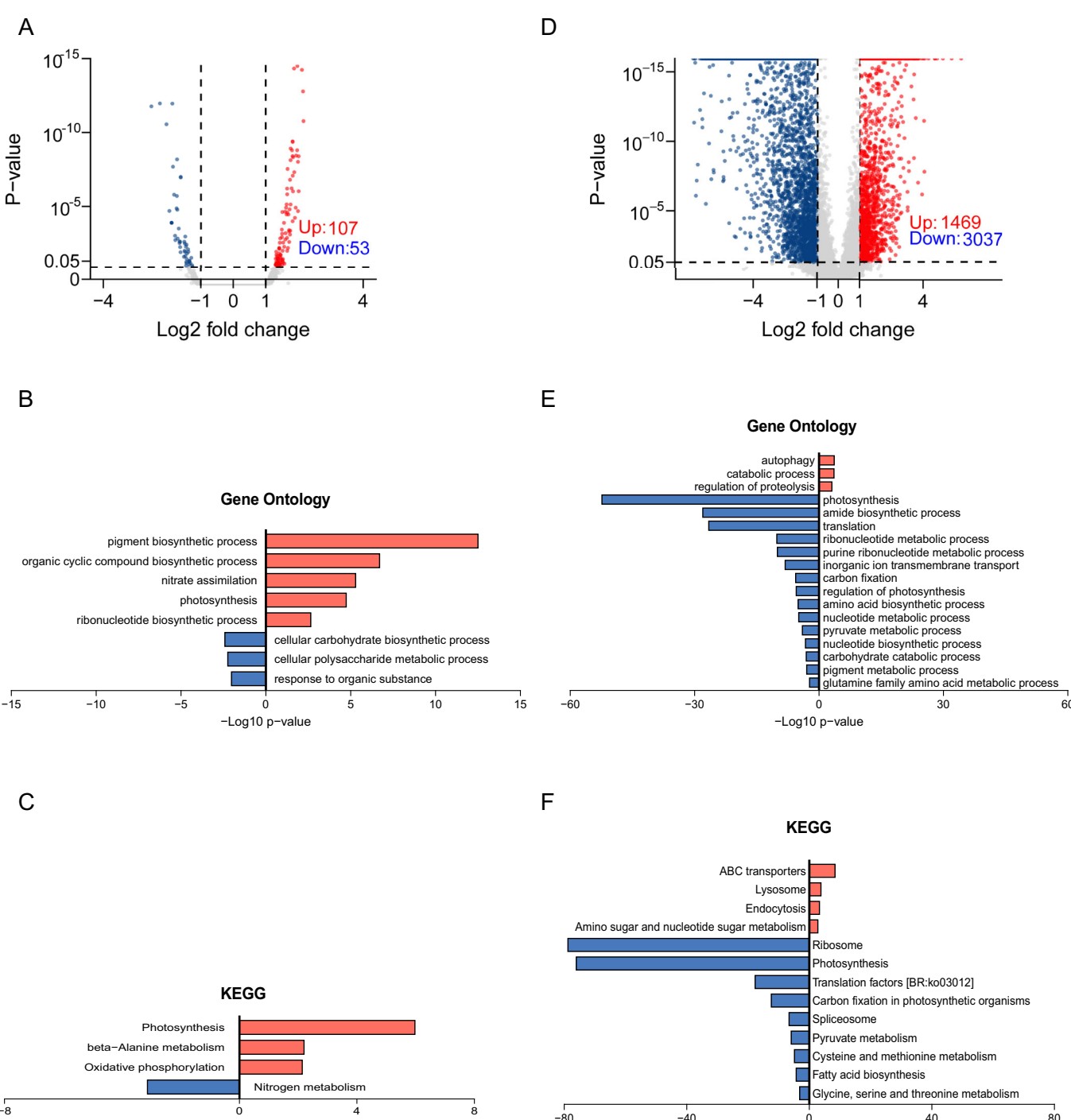

**Figure EV4.   N-depletion in the dinoflagellate _A. carterae_ induces minor differences in mRNA accumulation levels but dramatic changes in translation efficiency.**

(A) Volcano plot showing differentially expressed genes (DEGs) under N-depletion treatment. (B, C), GO and KEGG enrichment analysis revealing the enriched GO terms and pathways of these DEGs, respectively. (D) Volcano plot showing differential translation efficiency genes (DTEGs) after N-depletion treatment. (E, F) Functional analysis of these DTEGs, respectively. Red colors indicate either upregulated DEGs or DTEGs, while blue colors represent either downregulated DEGs or DTEGs. _p_-values were calculated using the Wald test (A), Hypergeometric test (B, C, E, F) and Chi-Squared Test (D).

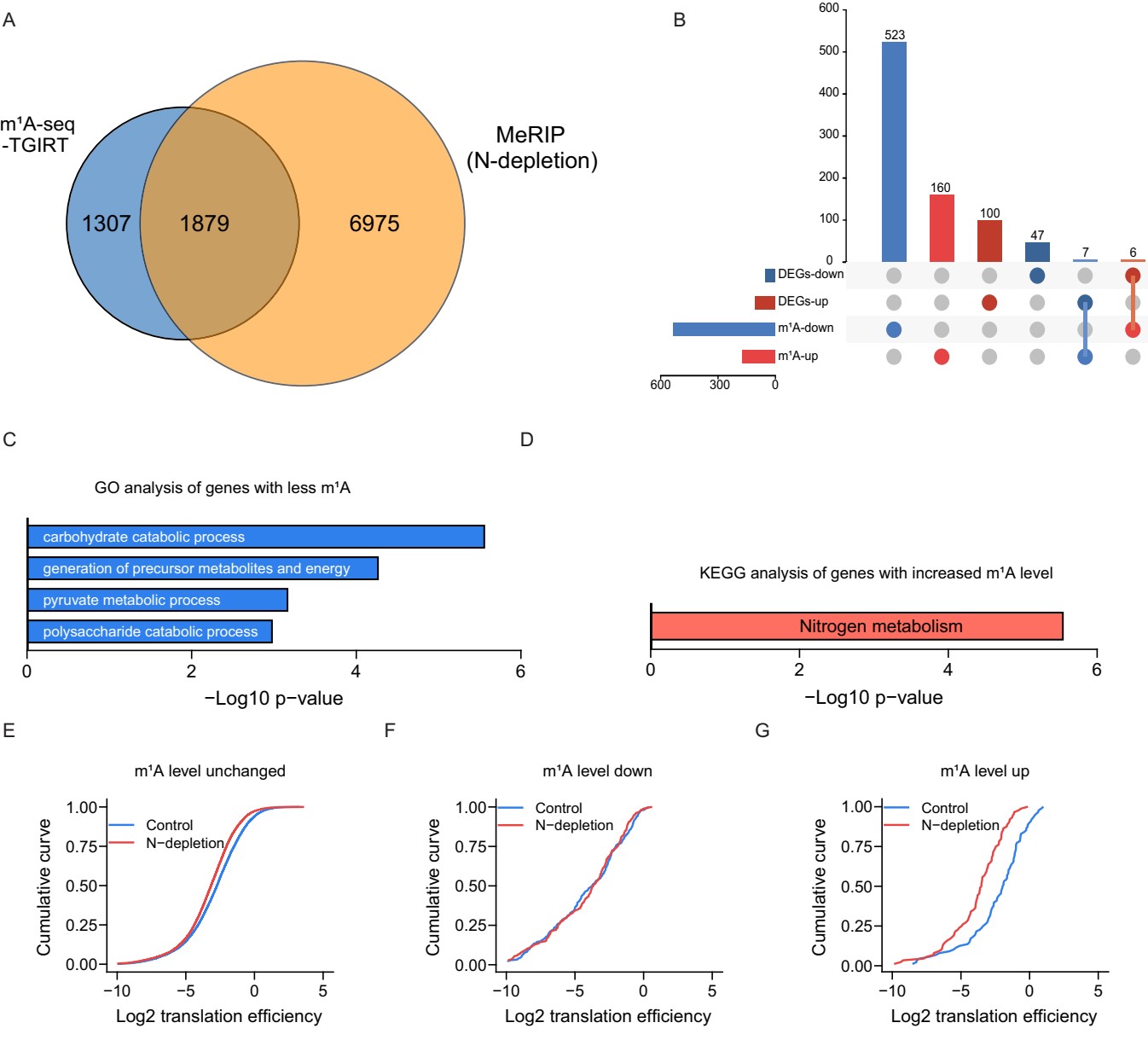

**Figure EV5.   Features of differentially methylated m¹A genes under N-depletion.**

(**A**) The Venn diagram depicting the overlap of genes identified as methylated by both m¹A-seq-TGIRT and MeRIP under N-depletion conditions. (**B**) The Upset plot showing the numbers of shared genes between DEGs and differentially methylated genes. (**C, D**) GO and KEGG enrichment analysis of the downregulated and upregulated m¹A-modified genes, respectively. *p*-values were calculated using hypergeometric test. (**E–G**) The translation efficiency is plotted as accumulative fractions for genes with unchanged methylation (**E**), decreased methylation (**F**), and elevated methylation (**G**) for both control and N-depletion treatment groups.

