## [Peer Review File · EMBO Reports]

Abundant mRNA m1A modification in dinoflagellates: a new layer of gene regulation

Chongping LI, Ying LI, Jia Guo, Yuci Wang, Xiaoyan SHI, Yangyi ZHANG, Nan LIANG, Honghui MA, Jie YUAN, Jiawei xu, and Hao Chen

Corresponding author(s): Hao Chen (chenh7@sustech.edu.cn), Jie YUAN (yuanjie@szhospital.com), Jiawei xu (fccxujw@zzu.edu.cn)

Review Timeline:

Submission Date:	12th Jan 24
Editorial Decision:	6th Feb 24
Revision Received:	28th Jun 24
Editorial Decision:	26th Jul 24
Revision Received:	1st Aug 24
Accepted:	22nd Aug 24

Editor: Esther Schnapp

Transaction Report:

Dear Dr. Chen,

Thank you for the submission of your manuscript to EMBO reports. We have now received the full set of referee reports as well as additional comments from another expert, who only had time to briefly look at your ms.

As you will see, the referees acknowledge that your findings are potentially interesting and a good fit for our journal. However, both referees 1 and 2 as well as the additional expert point out that the method used to detect m1A in dinoflagellates is not sufficiently conclusive, and that orthogonal methods are required for the identification of m1A sites. I think all points raised by the referees are reasonable and should be addressed. Please let me know in case you disagree or have any comments, and we can discuss the exact revision requirements further, also in a video chat, if you like.

I would thus like to invite you to revise your manuscript with the understanding that the referee concerns must be fully addressed and their suggestions taken on board. Please address all referee concerns in a complete point-by-point response. Acceptance of the manuscript will depend on a positive outcome of a second round of review. It is EMBO reports policy to allow a single round of major revision only and acceptance or rejection of the manuscript will therefore depend on the completeness of your responses included in the next, final version of the manuscript.

We realize that it is difficult to revise to a specific deadline. In the interest of protecting the conceptual advance provided by the work, we recommend a revision within 3 months (8th May 2024). Please discuss the revision progress ahead of this time with the editor if you require more time to complete the revisions.

- 1) A data availability section providing access to data deposited in public databases is missing. If you have not deposited any data, please add a sentence to the data availability section that explains that.
- 2) Your manuscript contains statistics and error bars based on $n=2$. Please use scatter blots in these cases. No statistics should be calculated if $n=2$.

3) We replaced Supplementary Information with Expanded View (EV) Figures and Tables that are collapsible/expandable online. A maximum of 5 EV Figures can be typeset. EV Figures should be cited as 'Figure EV1, Figure EV2' etc... in the text and their respective legends should be included in the main text after the legends of regular figures.

5) a complete author checklist, which you can download from our author guidelines . Please insert information in the checklist that is also reflected in the manuscript. The completed author checklist will also be part of the RPF.

6) Please note that all corresponding authors are required to supply an ORCID ID for their name upon submission of a revised manuscript (). Please find instructions on how to link your ORCID ID to your account in our manuscript tracking system in our Author guidelines

7) Before submitting your revision, primary datasets produced in this study need to be deposited in an appropriate public database (see <https://www.embopress.org/page/journal/14693178/authorguide#datadeposition>). Please remember to provide a reviewer password if the datasets are not yet public. The accession numbers and database should be listed in a formal "Data Availability" section placed after Materials & Method (see also <https://www.embopress.org/page/journal/14693178/authorguide#datadeposition>). Please note that the Data Availability Section is restricted to new primary data that are part of this study. * Note - All links should resolve to a page where the data can be accessed. *
If your study has not produced novel datasets, please mention this fact in the Data Availability Section.

- the name of the statistical test used to generate error bars and P values,
- the number (n) of independent experiments (please specify technical or biological replicates) underlying each data point,
- the nature of the bars and error bars (s.d., s.e.m.),
- If the data are obtained from n {less than or equal to} 2, use scatter blots showing the individual data points.

I look forward to seeing a revised form of your manuscript when it is ready.

Kind regards,
Esther

Referee #1:

This is an interesting manuscript that shows that m1A is prevalent in dinoflagellates mRNA. Several studies have looked at m1A in eukaryotes, focusing on mammalian cell lines and initial studies reported high levels of m1A but subsequent work showed that these were likely to be various types of artifacts. M1A is instead very rare in mammalian mRNA. Plus, it is surprising and interesting that there is a eukaryote that appears to have high levels of m1A. The authors characterize the distribution of m1A and provide reasonable data that m1A is indeed present in this organism.

There are two concerns. First the mapping doesn't appear to be done correctly:

1. As the authors probably know, after the original reports from the He lab and the Yi lab that m1A is present in mammalian mRNA, the Schwartz lab repeated the analysis and specifically looked for m1A -induced mutations, which is the hallmark of the presence of m1A. These mutations were missing in the m1A-immunoprecipitated RNAs. This helped Schwartz demonstrate that the m1A was unlikely to be true m1A sites. In the other study by the Jaffrey lab, they measured m1A induced transcription stops. Mutations/stops are really important mutational signatures that should be used when measuring or detecting m1A sites. These signatures provide the -exact- location of m1A. Sadly, it appears that the authors are using peaks, which are known to be displaced downstream of m1A sites due to the ability of m1A to induce transcription termination. Unless I'm misunderstanding, it seems like the authors of this study have not use the proper mutational signatures to definitively identify m1A in a nucleotide-specific manner. They should be able to mine their data or repeat their analysis with a slightly different library preparation protocol to identify the true m1A sites. It's important to identify all the m1A sites with nucleotide resolution since the methods are well-known. This is a fixable problem, but needs to be done properly for this paper to be acceptable. Peaks are no longer a valid way to map modifications in 2024.

The second concern has to do with validation of the m1A sites, and the m1A motifs.

2. The m1A sites need to be both validated and stoichiometry needs to be measured/predicted. There several ways to do this. The m1A can induce reverse transcription stops depending on the polymerase and the buffer conditions. It will be important to show that putative m1A sites indeed have stops, and the fraction of stopping needs to be established, using standards, such as ribosomal RNA if they cannot obtain a synthesized standard. Mutations can be used as well.

In particular, it seems possible that the m1A sites in the coding sequence largely derived from nonspecific signals whereas m1A in the 3' UTR is more likely to be a real signal. They may find different false positive rates depending on where they are looking. It's not a problem if some of the sites are not real, but the author should have some sort of measurement of false positive, and this may be related to the location within the transcript body.

Other methods could include SCARLET/SCARPET.

3. The analysis of motifs needs to be a little bit more clear. What percent of all the m1A sites fall within one of the three recognized motifs? Is the rank order/prevalence of the different motifs the same for the m1A sites in the 3' UTR as in the coding sequence? Or the 5' UTR? I suspect that the number of m1A sites that are found in the coding sequence that fall within one of the major motifs is much lower. However will be important to measure this.

Referee #2:

Prof. Hao Chen lab discovers abundant m1A methylations in mRNA of dinoflagellates. According to my knowledge of mRNA modifications, m1A methylation has been proven to be at a very low abundance in mRNA isolated from mammalian cells, which has hampered the functional investigation of its role in regulating mRNA metabolism. For nearly 7 years after the first characterization of m1A methylomes in human/mouse mRNA, people have been searching for a biological system of abundant mRNA m1A methylation. Here the authors have made great progress in demonstrating such a model system of dinoflagellates, and I believe its academic impact will trigger lots of future projects in studying mRNA m1A epigenetics. Here I support the publication of this paper in EMBO Reports, after minor revision to strengthen the bioinformatic part of this research.

(1) In Fig.2 and Fig.3, the authors conducted m1A-MeRIP-seq with m1A-specific antibody for mapping m1A profile in dinoflagellates mRNA. However, the analysis of RT mutation or truncation signatures is missing here, as we know that nearly all RT enzymes can induce mutation or truncation signals at m1A-modified sites. The in-depth analysis of RT mutation/truncation

signals at potential m1A sites will strengthen the discovery of m1A methylomes in dinoflagellates mRNA.

(2) In 'Input' libraries of m1A-MeRIP-seq, the authors may see low, moderate, and high mutation/truncation signals at m1A-modified sites. Then, in 'IP' libraries, the mutation/truncation ratios at these candidate sites should be elevated after m1A enrichment. This elevation in mutation/truncation at m1A sites (in IP vs. Input) could be utilized for validating the newly identified m1A methylated sites. Meanwhile, in 'Input' libraries, the mutation/truncation ratios at m1A sites could be used for the estimation of m1A methylation fraction, serving for m1A quantification.

Referee #3:

This manuscript describes detection and analysis of the role of an M1A modification in dinoflagellate mRNA. As such, this is interesting, as dinoflagellates seem to favor translation over transcription as a means of controlling gene expression, and modification of RNA may contribute to this.

There are a few issues that I was not able to fully understand and that should be made clear before the manuscript is accepted for publication. My biggest issue lies in measurements of translation efficiencies. I can readily believe that comparing the number of ribosome protected fragments for the same transcript between different conditions is informative, but I am less convinced that two different transcripts can be meaningfully compared. If this is really what is being done, it should be justified in the text.

Figure 1 reports that about 3% of adenines are modified in mRNA from *Amphidinium* (line 145). This means that a 1,000 base mRNA would have about 7 modified adenines (excluding the poly A tail). How does this correspond to a total of 123481 "peaks" in mRNAs of 10794 (line 168)?

Figure 2 reports a high number of genes with 1 peak. What is this "peak"? Me-RIP should identify sequences (average length about 100 nucleotides) that are precipitated by the anti m1A antibody. Does a peak thus correspond to a single region in an RNA molecule where the majority of the precipitated reads map to? This would correspond to panel C (a peak mostly in the 3'UTR) but does not correspond to what is presented in panel D. What is the vertical axis in panel D? The legend should also say the schema at the bottom of panel D is the genome sequence and the arrow is the direction of transcription. The motif GCCACGC (line 183) is not in Figure 2E. Lastly, how many non-methylated transcripts are there? It seems 10794 transcripts are methylated (line 168), how many transcripts were detected (i.e. what fraction of transcripts are methylated)?

Figure 3 compares methylated and non-methylated transcripts. Panel A indicates the non-methylated transcripts are less abundant. Is this because the probability of detecting a methylated RNA fragment is less? Panel B has gene expression as the x axis, I think this should read RNA abundance. Most of the m1A peaks should be found in the 3'UTR, does placing them elsewhere affect translation more? What is being measured in panel F? If this is derived from the number of ribosome protected fragment reads, I do not see how different transcripts can be meaningfully compared.

Up regulated genes measured by RNA-Seq include nitrate metabolism and photosynthesis. I would have expected photosynthesis to be reduced in the absence of nitrate as it should be more difficult to store the reduced carbon. However, this does seem to be better reflected by the ribosome protected fragment data.

In Figure 4 A, what does it mean that m1A methylation changes after N-depletion? That the number of methylated MeRIP fragments in a sequence decreases? Or the number of methylated sequences decreases? If the later then only a fraction of the transcripts of a given sequence are methylated? Panel F also seems to be comparing translation efficiency between different transcripts, which would pose a problem similar to that in figure 3F.

Additional comments from an expert:

The key claim in this manuscript is widespread m1A in Dinoflagellates. This would be a remarkable discovery, certainly worthy of publication in EMBO Reports. m1A is present only at trace levels in human mRNA, and hence such widespread deposition on mRNA would open up a plethora of questions regarding role, function, biogenesis and evolution.

My major concern is about the methodology used for detecting m1A. The authors begin their study by pointing out the so-called "controversiality" about m1A distribution in mammalian mRNA, a mark that was found to be widespread by initial studies but in follow-ups found to be extremely rare and low level. The divergence in the findings between the initial studies and the later ones is not due to differences in opinion or in implementation, but due to differences in methodology. The early studies relied on antibody based detection, whereas the later ones relied on misincorporation-based detection of m1A. As became apparent in the later studies, antibodies are prone to promiscuous binding, and the vast majority of sites reported in the early studies are thought to have been false hits. I was therefore disappointed that despite these severe limitations, the authors chose to map m1A sites using an antibody based approach. Given how central the discovery of m1A is to the message of this manuscript, in my opinion it would be critical that the authors confirm their findings on the basis of previously developed misincorporation based

approaches, including also enzymatic (alk) or chemical (Dimroth) elimination of m1A sites as an additional control. This point is all the more critical as the authors are unable to tie the m1A sites that they detect to a motif known to harbor m1A, or to an enzyme that deposits it, nor do they have single nucleotide resolution allowing them to confirm the sites using orthogonal approaches.

Key: Reviewer's comments in black; Author's response in blue; New or updated figures in red.

We thank all the reviewers for their comments on the importance of our study and their constructive criticisms, which we have addressed in full below.

Reviewers' Comments:

Referee #1:

This is an interesting manuscript that shows that m1A is prevalent in dinoflagellates mRNA. Several studies have looked at m1A in eukaryotes, focusing on mammalian cell lines and initial studies reported high levels of m1A but subsequent work showed that these were likely to be various types of artifacts. M1A is instead very rare in mammalian mRNA. Plus, it is surprising and interesting that there is a eukaryote that appears to have high levels of m1A. The authors characterize the distribution of m1A and provide reasonable data that m1A is indeed present in this organism.

Response: We thank this reviewer for recognizing the importance of our study and the constructive suggestions. Based on the reviewer's great suggestions, we have made extensive efforts to strengthen our study with additional data. Our point-by-point responses are included below.

There are two concerns. First the mapping doesn't appear to be done correctly:

1. As the authors probably know, after the original reports from the He lab and the Yi lab that m1A is present in mammalian mRNA, the Schwartz lab repeated the analysis and specifically looked for m1A -induced mutations, which is the hallmark of the presence of m1A. These mutations were missing in the m1A-immunoprecipitated RNAs. This helped Schwartz demonstrate that the m1A was unlikely to be true m1A sites. In the other study by the Jaffrey lab, they measured m1A induced transcription stops. Mutations/stops are really important mutational signatures that should be used when measuring or detecting m1A sites. These signatures provide the -exact- location of m1A. Sadly, it appears that the authors are using peaks, which are known to be displaced downstream of m1A sites due to the ability of m1A to induce transcription termination. Unless I'm misunderstanding, it seems like the authors of this study have not use the proper mutational signatures to definitively identify m1A in a nucleotide-specific manner. They should be able to mine their data or repeat their analysis with a slightly different library preparation protocol to identify the true m1A sites. It's important to identify all the m1A sites with nucleotide resolution since the methods are well-known. This is a fixable problem, but needs to be done properly for this paper to be acceptable. Peaks are no longer a valid way to map modifications in 2024.

Response: We thank the reviewer for this great suggestion. Per the reviewer's advice, we used misincorporation as the signature of m¹A sites and conducted m¹A-seq-TGIRT, following the procedure outlined in Schwartz lab's study (Safra et al, 2017), as detailed

in material and method section of our revised manuscript. Our new results indicated that the IP group has a higher global misincorporation ratio at A sites in mRNA compared to the input samples (new Fig. EV1B), suggesting that our newly identified m¹A sites are reliable. Based on single-base resolution of all identified m¹A sites, our new results still demonstrated that the majority of the m¹A sites are localized in 3'UTR of the dinoflagellate mRNA. Using this new strategy, we identified 6,549 high-confidence m¹A sites in the mRNAs of 3,196 genes (new Fig. 2A). Consistently, our new analysis still revealed that the presence of m¹A sites appears to negatively affect the translation efficiency of the m¹A-modified genes. The details of our results can be seen in the updated Fig. 2, Fig. 3 and Fig. EV3 of the revised manuscript.

The second concern has to do with validation of the m¹A sites, and the m¹A motifs.

2. The m¹A sites need to be both validated and stoichiometry needs to be measured/predicted. There several ways to do this. The m¹A can induce reverse transcription stops depending on the polymerase and the buffer conditions. It will be important to show that putative m¹A sites indeed have stops, and the fraction of stopping needs to be established, using standards, such as ribosomal RNA if they cannot obtain a synthesized standard. Mutations can be used as well.

In particular, it seems possible that the m¹A sites in the coding sequence largely derived from nonspecific signals whereas m¹A in the 3' UTR is more likely to be a real signal. They may find different false positive rates depending on where they are looking. It's not a problem if some of the sites are not real, but the author should have some sort of measurement of false positive, and this may be related to the location within the transcript body.

Other methods could include SCARLET/SCARPET.

Response: We appreciate the reviewer's comment. As mentioned in response to point #1, we used the m¹A-mediated mutation/misincorporation signature via the m¹A-seq-TGIRT approach to identify m¹A sites at base resolution. This method significantly reduced the false-positive m¹A-enriched peaks previously obtained through the m¹A-MeRIP method (Fig. EV1D).

Moreover, our sequencing results revealed some reads mapped to the large ribosomal subunit 28S rRNA. Analysis of conserved m¹A sites in the 28S rRNA of *Amphidinium carterae* revealed that the m¹A site in the sequence context (GAAACACGGACCAAGGAG, with the m¹A site highlighted in red), present in both mammalian and yeast cells (Sharma et al, 2018), exhibited **apparent misincorporation in both IP and input samples using m¹A-seq-TGIRT approach** (Fig. EV1C), further confirming the reliability of our newly identified m¹A sites. Our new results indicated that most of the m¹A sites are located in the 3'UTR of the encoded transcripts, while only 14.2% are in the CDS and 4.4% in the 5'UTR (new Fig. 2B).

3. The analysis of motifs needs to be a little bit more clear. What percent of all the m¹A sites fall within one of the three recognized motifs? Is the rank order/prevalence of the different motifs the same for the m¹A sites in the 3' UTR as in the coding sequence? Or

the 5' UTR? I suspect that the number of m¹A sites that are found in the coding sequence that fall within one of the major motifs is much lower. However will be important to measure this.

Response: We are grateful for this comment. We have updated the motif analysis of m¹A sites based on single-base resolution mapping, as shown in the new Fig. 2E, using combined upstream 3-mer distribution frequency analysis and sequence logo plotting. The results showed that m¹A is enriched in the NNCA sequence context of dinoflagellate mRNA (new Fig. 2E). Moreover, sequence logo plots of m¹A sites in different regions showed that the NNCA signature motif is also present in m¹A sites within the 5'UTR (46.21% with NNCA), CDS (44.27% with NNCA), and 3'UTR (49.16% with NNCA) (Figure for Review Only).

Figure for Review Only. The sequence logo frequency of m¹A sites identified by m¹A-seq-TGIRT occurring in different mRNA segments, including the 5'UTR (A), CDS (B), and 3'UTR (C).

Referee #2:

Prof. Hao Chen lab discovers abundant m¹A methylations in mRNA of dinoflagellates. According to my knowledge of mRNA modifications, m¹A methylation has been proven to be at a very low abundance in mRNA isolated from mammalian cells, which has hampered the functional investigation of its role in regulating mRNA metabolism. For nearly 7 years after the first characterization of m¹A methylomes in human/mouse mRNA, people have been searching for a biological system of abundant mRNA m¹A methylation. Here the authors have made great progress in demonstrating such a model system of dinoflagellates, and I believe its academic impact will trigger lots of future projects in studying mRNA m¹A epigenetics. Here I support the publication of this paper in EMBO Reports, after minor revision to strengthen the bioinformatic part of this research.

Response: We thank this reviewer for recognizing the importance of our study and the constructive suggestions.

(1) In Fig.2 and Fig.3, the authors conducted m¹A-MeRIP-seq with m¹A-specific antibody for mapping m¹A profile in dinoflagellates mRNA. However, the analysis of RT mutation or truncation signatures is missing here, as we know that nearly all RT enzymes can induce mutation or truncation signals at m¹A-modified sites. The in-depth analysis of RT mutation/truncation signals at potential m¹A sites will strengthen the discovery of m¹A methylomes in dinoflagellates mRNA.

Response: We appreciate the reviewer for this comment. As mentioned in response to points #1 and #2 of the Reviewer #1, we identified m¹A sites by using misincorporation as the signature of m¹A sites in our revised manuscript through performing m¹A TGIRT-seq. We could detect m¹A-mediated misincorporation in both input and IP samples using TGIRT enzyme (revised Fig. 2D and Fig. EV1B). With this new approach, we uncovered 6,549 m¹A sites distributed among the mRNAs of 3,196 genes (new Fig. 2A and 2B). As shown in new Fig. 2 and Fig. 3, we also re-analyzed the m¹A-modified genes identified by single-base resolution sequencing.

(2) In 'Input' libraries of m¹A-MeRIP-seq, the authors may see low, moderate, and high mutation/truncation signals at m¹A-modified sites. Then, in 'IP' libraries, the mutation/truncation ratios at these candidate sites should be elevated after m¹A enrichment. This elevation in mutation/truncation at m¹A sites (in IP vs. Input) could be utilized for validating the newly identified m¹A methylated sites. Meanwhile, in 'Input' libraries, the mutation/truncation ratios at m¹A sites could be used for the estimation of m¹A methylation fraction, serving for m¹A quantification.

Response: We thank the reviewer for this great advice. As mentioned in responses to points #1 and #2 of the Reviewer #1, through the m¹A-seq-TGIRT strategy, we could detect various levels of mutation ratios at putative m¹A-modified sites in the input samples (Fig. EV1B). Consistently, we indeed observed a significantly increase of mutation ratio in the IP group than the inputs (Fig. EV1B).

Referee #3:

This manuscript describes detection and analysis of the role of an M¹A modification in dinoflagellate mRNA. As such, this is interesting, as dinoflagellates seem to favor translation over transcription as a means of controlling gene expression, and modification of RNA may contribute to this.

Response: We thank this reviewer for recognizing the importance of our study and the constructive suggestions.

There are a few issues that I was not able to fully understand and that should be made clear before the manuscript is accepted for publication. My biggest issue lies in measurements of translation efficiencies. I can readily believe that comparing the number of ribosome protected fragments for the same transcript between different conditions is informative, but I am less convinced that two different transcripts can be meaningfully compared. If this is really what is being done, it should be justified in the text.

Response: We appreciate the reviewer's concern. We agree with the reviewer that comparing the translation efficiencies of different transcripts cannot simply rely on counting the ribosome protected fragments (RPF), the level of which is also significantly affected by the abundance of mRNA/transcripts. In our analysis, we used TE (Translation Efficiency) as a parameter to compare the translation of different mRNA/transcripts. The TE was calculated through dividing the normalized RPF counts by mRNA abundance (TE = normalized RPF counts/mRNA abundance).

Similar comparison of TE for different transcripts have been widely utilized to investigate the regulation of mRNA translation by RNA modifications, including m⁶A (Mao et al, 2019) and ac⁴C (Arango et al, 2018). More recently, the TE of different transcripts was also utilized as a signature to identify the 5'UTR candidates with optimal mRNA translation (Cao et al, 2021). Therefore, we believe that our comparison of different transcripts' translation via measuring TE is a reliable way to investigate the roles of m¹A in controlling mRNA translation. Per the reviewer's advice, we included more detailed information about TE calculation in the "Bioinformatics analyses of RNA-seq, Ribo-seq, MeRIP-seq and m¹A-seq-TGIRT data" subsection of the Methods section in the revised manuscript to avoid the potential misunderstanding for readers.

Figure 1 reports that about 3% of adenines are modified in mRNA from Amphidinium (line 145). This means that a 1,000 base mRNA would have about 7 modified adenines (excluding the poly A tail). How does this correspond to a total of 123481 "peaks" in mRNAs of 10794 (line 168)?

Response: We thank the reviewer for raising this issue. Basically, these are two completely different experiments and the numbers from them cannot be directly compared. The m¹A modification ratio (~3% of the total adenines) is determined using LC-MS/MS, which requires enzyme digestion into mono-nucleosides and thus cannot provide sequence information regarding the methylated fragments of the transcripts. However, the number of peaks is detected through high-throughput sequencing of m¹A-modified mRNA fragments using an anti-m¹A antibody. Each "peak" identified in the MeRIP-seq analysis represents a region within the mRNA that is enriched for m¹A modifications. Given that a single transcript can have multiple m¹A modification sites, multiple peaks can be identified within a single mRNA molecule.

Figure 2 reports a high number of genes with 1 peak. What is this "peak"? Me-RIP should identify sequences (average length about 100 nucleotides) that are precipitated by the anti m¹A antibody. Does a peak thus correspond to a single region in an RNA molecule where the majority of the precipitated reads map to? This would correspond to panel C (a peak mostly in the 3'UTR) but does not correspond to what is presented in panel D. What is the vertical axis in panel D? The legend should also say the schema at the bottom of panel D is the genome sequence and the arrow is the direction of transcription. The motif GCCACGC (line 183) is not in Figure 2E. Lastly, how many non-methylated transcripts are there? It seems 10794 transcripts are methylated (line 168), how many transcripts were detected (i.e. what fraction of transcripts are methylated)?

Response: We thank the reviewer for this comment. Because of the non-specific binding of the immunoprecipitation, we need to subtract the background signal using a control sample (input) to identify the potential m¹A-enriched peaks. These peaks indicate regions where the anti-m¹A antibody has successfully precipitated RNA fragments, showing significant enrichment of m¹A modifications. Panel C in the previous Fig. 2 presents a metagene distribution analysis of all identified peaks, highlighting their

locations along different transcript regions.

Panel D in the previous Fig. 2 showed examples of two m¹A peaks (highlighted by the pale-yellow color and the arrow) in separate transcripts. The vertical axis in Panel D represents the normalized sequencing read coverage, with higher values indicating greater read coverage.

Since we can detect m¹A sites at single-base resolution now, the distribution pattern and motifs of m¹A sites are more accurate. Fig. 2 was also updated in the revised manuscript accordingly. As shown in the updated Fig. 2E in the revised manuscript, the combined upstream 3-mer distribution frequency analysis and sequence logo plot suggest that m¹A in dinoflagellate mRNA is enriched in the NNCA sequence context. Additionally, we apologize for the typographical error of the motif "GCCACGC" (line 183) in the previous version of the manuscript, which had been corrected in the revised manuscript.

To the last question, the number of transcripts is largely based on the cutoff of TPM (transcripts per million) for transcripts that can be considered as “expressed”. In current study, we used a threshold of TPM > 5 and identified a total of 34574 transcripts, among which 10794 transcripts (~ 31.2% of total) were detected to be m¹A-modified through m¹A-RIP-seq. To avoid the false-positive sites caused by the anti-m¹A antibody, we conducted m¹A-seq-TGIRT for single-base resolution analysis of m¹A sites in the *A. carterae* transcriptome in the revised manuscript. This new method detected 6549 m¹A sites in the mRNAs of 3196 genes (~ 9.2% of total) under normal growth condition.

Figure 3 compares methylated and non-methylated transcripts. Panel A indicates the non-methylated transcripts are less abundant. Is this because the probability of detecting a methylated RNA fragment is less? Panel B has gene expression as the x axis, I think this should read RNA abundance. Most of the m¹A peaks should be found in the 3'UTR, does placing them elsewhere affect translation more? What is being measured in panel F? If this is derived from the number of ribosome protected fragment reads, I do not see how different transcripts can be meaningfully compared.

Response: We thank the reviewer for these great comments. The response to every specific question had been listed as below. (Note: all the results in the Fig. 3 had also been updated with the m¹A-modified genes identified through single-base resolution analysis.)

Panel A: We thank the reviewer for this comment. We guess that this probability would be very low due to these following reasons: 1) When we searched for the m¹A peaks, we used the value of m¹A-enrichment fold (IP/Input: m¹A IP counts normalized by RNA abundance from Input) of every individual region. This would effectively minimize the influences related with RNA abundance; 2) To ensure the similar detectability of all compared transcripts, we only included the transcripts of considerable abundance (TPM > 5) for the comparison in our analysis.

Panel B: We thank the reviewer for pointing this out. We have corrected the label of x axis into mRNA abundance. We agree with reviewer's conclusion on the effects of m¹A in different regions. As shown in our new Fig. 3H, m¹A sites in CDS regions appear to have a greater impact on translation efficiency than those in the 3'UTR, implying a regional effect of m¹A modification.

Panel F: Panel F demonstrated that the translation efficiency of genes between two categories (m¹A vs non-m¹A) was compared. As the previous responses to the general comments, we used TE (Translation Efficiency) as a parameter to compare the translation of different mRNA/transcripts, following the standard protocol widely used to investigate the regulation of mRNA translation by RNA modifications. More details can be found in the reply to the general comments.

Up regulated genes measured by RNA-Seq include nitrate metabolism and photosynthesis. I would have expected photosynthesis to be reduced in the absence of nitrate as it should be more difficult to store the reduced carbon. However, this does seem to be better reflected by the ribosome protected fragment data.

Response: We agree with the reviewer on this point. Dinoflagellate gene expression is believed to mainly dependent on pos-transcriptional regulation (Roy and Morse, 2013; Zaheri and Morse, 2022). Indeed, it seems that translation control also plays more important roles compared with transcription regulation in dinoflagellate *A. carterae*. We have included this piece of information into a sentence of the revised manuscript “With an increase in photosynthesis at the transcriptional level but a decrease at the translational level, and a much more significant change in translation efficiency compared to mRNA levels, our data demonstrate that the dinoflagellate *A. carterae* adapts to N-depletion conditions primarily by regulating mRNA translation rates.”

In Figure 4 A, what does it mean that m¹A methylation changes after N-depletion? That the number of methylated MeRIP fragments in a sequence decreases? Or the number of methylated sequences decreases? If the later then only a fraction of the transcripts of a given sequence are methylated? Panel F also seems to be comparing translation efficiency between different transcripts, which would pose a problem similar to that in figure 3F.

Response: We appreciate the reviewer's comments. Based on the question, we guess the reviewer was mentioning Fig. 4C rather than Fig. 4A. Initially, we measured the change in global m¹A methylation level in mRNA using LC/MS-MS techniques between *A. carterae* cells grown under normal and nitrogen-depletion (N-depletion) conditions. We found an approximately 30% reduction in the m¹A level under N-depletion growth conditions (Fig. 4C), indicating a decrease in the overall level of m¹A methylation. This result only reflected the global changes of m¹A modification. It's difficult to postulate that the decrease is caused by reduction in number or methylation level of methylated sequences. To answer this question, we had to resort to next-generation sequencing. Subsequently, we used MeRIP-seq to identify the differentially methylated regions of transcripts under N-depletion conditions. The results demonstrated that at least the

number of methylated sequences decreased (Fig. 4D).

Fig. 4F: As mentioned above (responses concerning Fig 3), this comparison is valid and widely used in translation studies. In Figure 4F, we compared the translation efficiency of different transcripts categorized by their methylation status. By comparing the overall translation efficiency between methylated and non-methylated genes, we could infer the impact of m¹A methylation on translation efficiency.

Note: Fig. 4D-4F had been updated with the newly identified m¹A sites, while the conclusion that m¹A plays an important role in regulating translation efficiency under N-depletion remains unchanged.

Additional comments from an expert:

The key claim in this manuscript is widespread m¹A in Dinoflagellates. This would be a remarkable discovery, certainly worthy of publication in EMBO Reports. m¹A is present only at trace levels in human mRNA, and hence such widespread deposition on mRNA would open up a plethora of questions regarding role, function, biogenesis and evolution.

My major concern is about the methodology used for detecting m¹A. The authors begin their study by pointing out the so-called "controversiality" about m¹A distribution in mammalian mRNA, a mark that was found to be widespread by initial studies but in follow-ups found to be extremely rare and low level. The divergence in the findings between the initial studies and the later ones is not due to differences in opinion or in implementation, but due to differences in methodology. The early studies relied on antibody based detection, whereas the later ones relied on misincorporation-based detection of m¹A. As became apparent in the later studies, antibodies are prone to promiscuous binding, and the vast majority of sites reported in the early studies are thought to have been false hits. I was therefore disappointed that despite these severe limitations, the authors chose to map m¹A sites using an antibody based approach. Given how central the discovery of m¹A is to the message of this manuscript, in my opinion it would be critical that the authors confirm their findings on the basis of previously developed misincorporation based approaches, including also enzymatic (alk) or chemical (Dimroth) elimination of m¹A sites as an additional control. This point is all the more critical as the authors are unable to tie the m¹A sites that they detect to a motif known to harbor m¹A, or to an enzyme that deposits it, nor do they have single nucleotide resolution allowing them to confirm the sites using orthogonal approaches.

Response: We thank this expert for recognizing the importance of our study and the constructive suggestions. We carried out extensive revision to our previous manuscript following this comment. The details can be found in the responses to reviewer #1 & #2.

References

Arango D, Sturgill D, Alhusaini N, Dillman AA, Sweet TJ, Hanson G, Hosogane M,

Sinclair WR, Nanan KK, Mandler MD et al (2018) Acetylation of cytidine in mRNA promotes translation efficiency. *Cell* 175: 1872-1886.e1824

Cao J, Novoa EM, Zhang Z, Chen WCW, Liu D, Choi GCG, Wong ASL, Wehrspaun C, Kellis M, Lu TK (2021) High-throughput 5' UTR engineering for enhanced protein production in non-viral gene therapies. *Nat Commun* 12: 4138

Mao Y, Dong L, Liu X-M, Guo J, Ma H, Shen B, Qian S-B (2019) m⁶A in mRNA coding regions promotes translation via the RNA helicase-containing YTHDC2. *Nat Commun* 10: 5332

Roy S, Morse D (2013) Transcription and maturation of mRNA in Dinoflagellates. *Microorganisms* 1: 71-99

Safra M, Sas-Chen A, Nir R, Winkler R, Nachshon A, Bar-Yaacov D, Erlacher M, Rossmannith W, Stern-Ginossar N, Schwartz S (2017) The m¹A landscape on cytosolic and mitochondrial mRNA at single-base resolution. *Nature* 551: 251-255

Sharma S, Hartmann JD, Watzinger P, Klepper A, Peifer C, Kötter P, Lafontaine DLJ, Entian K-D (2018) A single N1-methyladenosine on the large ribosomal subunit rRNA impacts locally its structure and the translation of key metabolic enzymes. *Sci Rep* 8: 11904

Zaheri B, Morse D (2022) An overview of transcription in dinoflagellates. *Gene* 829: 146505

Dear Dr. Chen,

Thank you for the submission of your revised manuscript. We have now received the enclosed reports from the referees that were asked to assess it, and I am happy to say that all support its publication now.

Only a few editorial requests will need to be addressed before we can proceed with the official acceptance of your manuscript:

- Please correct the conflict of interest subheading to "Disclosure and Competing Interests Statement"
- We need an institutional email address for the co-corresponding author Jiawei Xu.
- Please remove the author credits from the ms file. All author contributions need to be entered in our online ms submission system.
- Please delete the statement "data not shown" from page 15 as per journal policy.
- The following info also needs to be entered in our online ms submission system as separate Funders: National Natural Science Foundation of China (Grant Nos. 32170819 and 32170604), Pearl River Recruitment Program of Talents (2021QN02Y122), Department of Health of Guangdong Province (B2021032), Shenzhen Key Laboratory of Gene Regulation and Systems Biology (Grant No. ZDSYS20200811144002008) from Shenzhen Innovation Committee of Science and Technology and Funding for Scientific Research, the Scientific and Technological Innovation Team Project of Universities (Grant No. 24IRTSTHN037) and the Joint Fund for the Cultivation of Superior Disciplines of Henan Province (Grant No. 222301420013), the Medical Appropriate Technology Promotion project, the Young and Middle-aged Academic Leaders and the Leading Talents of Henan Health Commission, and Funding for Scientific Research and Innovation Team of The First Affiliated Hospital of Zhengzhou University (ZYCXTD2023004).
- Please upload all main and all EV figures as individual, high quality figure files.
- Please add a callout for Fig. 2A to the ms text.
- The Appendix file needs page numbers in the table of content on the title page; and each figure legend needs to be provided directly after its figure
- The Reagent and tools table is missing in the methods section. Effective 1 July 2024, all research articles submitted as revised versions must include a structured methods section that includes a Reagents and Tools Table followed by a Methods and Protocols section. You can find more information about this in our guide to authors online: <https://www.embopress.org/page/journal/14693178/authorguide#structuredmethods>
- The SOURCE DATA needs to be provided with a completed checklist that you should have received from Hannah Sonntag, and it needs to be uploaded as one (zipped) folder per figure.
- The manuscript sections should be in the following order: Title page - Abstract & Keywords - Introduction - Results & Discussion - Methods - Data Availability - Acknowledgments - Disclosure Statement & Competing Interests - References - Figure Legends - (Main Tables with legends) - Expanded View Figure Legends.
- Please add the specific URL for the GSE246953 dataset to the data availability section.
- Please note that the exact p values are not provided in the legends of figures 3a, c-f; 4c, f; EV 3a, d.
- Please indicate the statistical test used for data analysis in the legends of figures EV 1e-f; EV 3a; EV 4a-f; EV 5c-d.
- Please note that information related to n is missing in the legends of figures 3a, c-f; 4a; EV 3b.
- Although 'n' is provided, please describe the nature of entity for 'n' in the legend of figure 4c.
- Please note that the error bars are not defined in the legend of figure 4a.

I would like to suggest a few changes to the title and abstract that needs to be written in present tense. Please let me know whether you agree with the following:

Abundant mRNA m1A modification in dinoflagellates: a new layer of gene regulation

Dinoflagellates, a class of unicellular eukaryotic phytoplankton, exhibit minimal transcriptional regulation, representing a unique model for exploring gene expression. The biosynthesis, distribution, regulation, and function of mRNA N1-methyladenosine (m1A) remain controversial due to its limited presence in typical eukaryotic mRNA. This study provides a comprehensive map of m1A in dinoflagellate mRNA and shows that m1A, rather than N6-methyladenosine (m6A), is the most prevalent internal mRNA modification in various dinoflagellate species, with an asymmetric distribution along mature transcripts. In *Amphidinium carterae*, we identify 6549 m1A sites characterized by a non-tRNA T-loop-like sequence motif within the transcripts of 3196 genes, many of which are involved in regulating carbon and nitrogen metabolism. Enriched within 3'UTRs, dinoflagellate mRNA m1A levels negatively correlate with translation efficiency. Nitrogen depletion further decreases mRNA m1A levels. Our data suggest that distinctive patterns of m1A modification influence the expression of metabolism-related genes through translational control.

EMBO press papers are accompanied online by A) a short (1-2 sentences) summary of the findings and their significance, B) 2-3 bullet points highlighting key results and C) a synopsis image that is exactly 550 pixels wide and 200-600 pixels high (the height is variable). The synopsis image should provide a sketch of the major findings, like a graphical abstract. Please note that text needs to be readable at the final size. Please send us this information along with the final manuscript.

Referee #1:

Overall this is a well executed study that makes the surprising finding that m1A (a modified nucleotide) is widely abundant in mRNA from dinoflagellates. m1A was claimed to be high in mammalian mRNA but two studies (Grozhiik et al and Schwartz et al.) showed the maps were artifacts of a bad antibody. This study uses the method developed by Schwartz (m1A-TIGRT-seq) and definitively shows m1A in dinoflagellates mRNA, which is very surprising and unusual. They test a candidate writer, and show it is not mediating m1A, so the enzyme writer is currently unknown. They provide evidence of a consensus site and a plausible function (mRNA translation inhibition likely due to stalling at m1A). Overall well done - I have no concerns.

Referee #2:

The revised version has addressed all my questions. I support the publication of this manuscript in EMBO reports.

Referee #3:

The authors have adequately addressed all my previous concerns.

All editorial and formatting issues were resolved by the authors.

Dr. Hao Chen
SUSTech
1088 Xueyuan Avenue
Shenzhen, Guangdong 518055
China

Dear Dr. Chen,

I am very pleased to accept your manuscript for publication in the next available issue of EMBO reports. Thank you for your contribution to our journal.
